# Tight Lower Bounds on Worst-Case Guarantees for Zero-Shot Learning with Attributes

**Alessio Mazzetto**\*
Brown University

**Cristina Menghini**\*
Brown University

**Andrew Yuan**
Brown University

**Eli Upfal**
Brown University

**Stephen H. Bach**
Brown University

## Abstract

We develop a rigorous mathematical analysis of zero-shot learning with attributes. In this setting, the goal is to label novel classes with no training data, only detectors for attributes and a description of how those attributes are correlated with the target classes, called the class-attribute matrix. We develop the first non-trivial lower bound on the worst-case error of the best map from attributes to classes for this setting, even with perfect attribute detectors. The lower bound characterizes the theoretical intrinsic difficulty of the zero-shot problem based on the available information—the class-attribute matrix—and the bound is practically computable from it. Our lower bound is tight, as we show that we can always find a randomized map from attributes to classes whose expected error is upper bounded by the value of the lower bound. We show that our analysis can be predictive of how standard zero-shot methods behave in practice, including which classes will likely be confused with others.

## 1 Introduction

Labeled training data is often scarce or unavailable, and it can be very costly to obtain. For this reason, there is a growing interest in developing methods that can exploit source of information other than labeled data, such as zero-shot learning (ZSL). In ZSL, we want to recognize items of *unseen classes*, for which labeled data is not available. A ZSL model is trained on a disjoint set of similar classes, called *seen classes*, for which labeled data is available instead. The model is trained to map examples to auxiliary information describing the seen classes. Then, at test time, predictions can be made using only descriptions of the unseen classes. While ZSL is increasingly common in practice, from a theoretical perspective ZSL is a hard problem that defies analysis, because in the worst case there can be an arbitrary shift between the distributions of the seen and unseen classes. In this work, we take a step towards a better theoretical understanding of ZSL. We investigate the question: *Given only auxiliary information in the form of attributes describing unseen classes, what is the smallest worst-case error than* **any** *method can guarantee?* We provide the first non-trivial answer to this question by developing a framework based on adversarial optimization. We also show that this framework has practical application as a method for identifying when the predictions of ZSL methods on certain unseen classes are more likely to be incorrect.

ZSL models have obtained impressive accuracy in practice, both for vision (Xian et al., 2018a) and language domains (Sanh et al., 2022; Wei et al., 2022), but they come with no theoretical

---

\* Equal contribution.

36th Conference on Neural Information Processing Systems (NeurIPS 2022).

characterization of their accuracy. To address this gap, we analyze the attribute-based ZSL setting that includes a large portion of the classic methods proposed in the literature (Romera-Paredes & Torr, 2015; Lampert et al., 2014; Akata et al., 2015, 2016), as well as more recent end-to-end deep learning approaches (Kodirov et al., 2017; Xian et al., 2018a; Huynh & Elhamifar, 2020). While this setting does not include all varieties of ZSL (discussed further in Section 2), we view this work as a critical first step towards building up a broader theory of ZSL. In attribute-based ZSL, an attribute is a property of a item to be classified. Each item can either exhibit a given attribute or not. For example, an image of a lion would often exhibit the attribute tail, while the image of a sheep would not. Attribute-based ZSL models are trained using *attribute representations* of the items of the seen classes, and a *class-attribute* matrix that describes pairwise relations between the seen classes and each attribute. At test time, predictions are made for the unseen classes given the items' attribute representation and a new class-attribute matrix.

Romera-Paredes & Torr (2015) is one of the few works to address theoretical questions related to ZSL. Studying attribute-based ZSL, they show a pair of basic bounds that characterize sufficient conditions for either learning or impossibility: (1) if there is no shift from the seen to the unseen classes, then learning is trivial, and (2) if the vectors of attributes of the seen classes are mutually orthogonal with those of the unseen classes, then the error can be arbitrarily large in the worst case. In this paper, we provide the first non-trivial lower bound for ZSL with attributes, addressing the open problem posed by Romera-Paredes & Torr (2015).

We analyze ZSL with attributes by first observing that it is a two stage process consisting of a training phase and an inference phase. In the training phase, we learn a map from the items to the attribute space using the seen classes, while in the inference phase we use the class-attribute matrix to infer the correct class given the item-attribute representation. Based on this two-stage decomposition, we can identify two kinds of errors. The first kind, related to the training phase, is due to domain shift. The map from items to the attribute space that is trained on the seen classes might not generalize accurately to the unseen classes. This contribution to the error can be arbitrarily large without introducing strong assumptions, as no labeled data is available for the unseen classes. Thus it is impossible to characterize the domain shift between seen and unseen classes. The second kind, related to the inference phase, is due to the fact that the class-attribute matrix might not fully differentiate among the unseen classes. In particular, there can be an item of the unseen classes with a set of attributes that according to the class-attribute matrix relation conforms with the description of two different classes. The first kind of error, domain shift, has been extensively studied both in the theory and the experimental literature (Mansour et al., 2009; Ben-David et al., 2010; Sener et al., 2016; Pinheiro, 2018; Luo et al., 2019). In this work, for the first time, we theoretically characterize the contribution of the second kind of error. It is important to understand and characterize this error for specific ZSL tasks because it corresponds to an inherent information gap in the problem setting that cannot be circumvented with a smarter algorithm.

We provide tight lower and upper bounds on the worst-case error of the best map from attribute representations to classes based on the class-attribute matrix. Our analysis gives a lower bound in the sense that it bounds from below the minimum error that any method can guarantee given only the information of the class-attribute matrix. The class-attribute matrix specifies the fraction of items in each class that exhibit each attribute. There is a range of class-attribute distributions that satisfy the constrains defined by a given matrix. We give a lower bound on the error of the best possible method for the worst case distribution in that range. This distribution represents a worst case correlation between attributes that satisfies the class-attribute matrix while maximizing the difficulty to distinguish between attribute-representation of items belonging to distinct classes. Our analysis also gives an upper bound in the sense that we show a randomized classifier that achieves at most the error of the lower bound, assuming perfect item-to-attribute mapping. This also shows that the lower bound is tight. Interestingly, the value of the lower bound can also be interpreted as the quality of the information provided by the class-attribute matrix. To the best of our knowledge, this is the first work to quantify such information.

**Contributions.** Our main contributions are the following:

1. We show the first non-trivial lower bound for attribute-based ZSL (Section 4).
2. We formulate the lower bound given a class-attribute matrix as a linear program (Section 4.1).
3. We show a closed form expression for the lower bound for binary classification (Section 4.2).

4. We show that the lower bound is tight: we exhibit a randomized classifier whose expected error is upper bounded by the value of the lower bound (Section 4.3).
5. We run extensive experiments comparing the theoretical results with the error of popular attribute-based ZSL methods, on benchmark datasets. We show that information given by the bound can be predictive of how standard methods behave, including which classes will likely be confused with others (Section 5).

## 2   Background and Related Work

Much early work on ZSL focused on using logical descriptions of the classes as auxiliary information, including attributes (Chang et al., 2008; Lampert et al., 2009). Since then, an increasing number of ZSL methods have been proposed, which differ in methodology and the auxiliary information they use. Examples of auxiliary information are symbolic descriptions of classes (Chang et al., 2008; Lampert et al., 2009), pre-trained embedding of the classes (Frome et al., 2013), natural language descriptions (Obeidat et al., 2019; Brown et al., 2020), and knowledge graphs (Wang et al., 2018; Kampffmeyer et al., 2019; Nayak & Bach, 2020). Recent ZSL methods can be grouped into two main categories: embedding-based and generation-based (Pourpanah et al., 2020). Seminal embedding-based works used two-layer neural networks to link the image feature space to the semantic one (Socher et al., 2013). Later, they evolved into deep neural networks that either map semantic features into the visual space (Ba et al., 2015; Zhang et al., 2017; Changpinyo et al., 2017) or project both the image and semantic features into the same space (Zhang & Saligrama, 2015; Radford et al., 2021). Generative-based approaches employ various kind of Generative Adversarial Networks (GANs) (Mirza & Osindero, 2014) to synthesize the features of the unseen classes, and use them to train a ZSL classifier in a supervised fashion (Felix et al., 2018; Li et al., 2019; Xian et al., 2018b, 2019; Narayan et al., 2020).

ZSL with attributes generally consists of learning a linear map from the item to the attribute space, in the first stage. Then, we use the class-attribute matrix to infer the correct class given the item-attribute representation (Xian et al., 2018a). ZSL with attributes can be seen as a special case of embedding-based ZSL, in which the class embeddings are the rows of the class-attribute matrix. Analyzing more general embedding-based or generation-based ZSL methods is challenging because they rely on deep neural networks for which relatively little theory is available.

Inspired by previous work to describe classes using error-correcting output codes (Dietterich & Bakiri, 1994), Palatucci et al. (2009) were the first to propose a ZSL algorithm for which they can provide a theoretical analysis. The algorithm learns linear classifiers individually for each binary attribute, and the attributes are mapped to the closest class-attribute representation. While they are able to provide a PAC bound, their analysis relies on several strong assumptions that limit the problem setting. First, they assume that they can learn each attribute independently, but attribute dependency is a widely recognized problem for attribute detection (Jakulin & Bratko, 2003). Second, they assume that each class has a unique attribute representation, i.e. each attribute must be either present or not in all the items of a given class. Finally, they also assume that they are able to sample classes from a given distribution, and they are able to generalize to the non-sampled classes. That is, they do not separate beforehand between seen and unseen classes, which is the common scenario observed in ZSL settings. Conversely, our lower bound does not assume a unique binary representation for each class, as we are given a class-attribute matrix that provides the probabilities to observe an attribute given an item of a class. Also, our lower bound takes into account the possible correlation between attributes, and it is computed based on the information provided on the given unseen classes.

In more recent work, Romera-Paredes & Torr (2015) draw a connection between transfer learning (Ben-David et al., 2010) and ZSL to provide a novel theoretical result. In particular, they show that their model is not able to generalize if the attribute representations of the seen classes are orthogonal to the one of the unseen classes. Intuitively, if those representations are orthogonal, the attribute map learned for the seen classes would fail to provide information for the unseen classes. This is an impossibility result, and it is not able to arbitrarily quantify the information given for the unseen classes. Unfortunately, transfer learning or domain adaptation like-bounds are challenging to estimate in a ZSL setting. In fact, a term of those bounds require access to labeled data for the unseen classes, which is unavailable in ZSL. Another term, the discrepancy, depends on the difference between the attribute representations of the classes and the distribution of the items between seen and unseen. While it would be theoretically possible to compute the discrepancy based on the

information available, its computation is very challenging and it has been possible only in very specific cases (Mansour et al., 2009).

Our novel lower bound is developed using adversarial techniques that describe the worst-case scenario with respect to the information available. It is inspired by recent work on semi-supervised learning, where the goal is to use the information provided by weak supervision sources (Balsubramani & Freund, 2015; Arachie & Huang, 2021; Mazzetto et al., 2021b;a). The adversarial approach allows us to handle the possible dependencies between the attributes.

## 3 Preliminaries

We denote scalar and generic items using lowercase letters, vectors using lowercase bold letters, and matrices using bold uppercase letters. Given two vectors $v$ and $v'$, we denote with $vv'$ the concatenation of the two vectors. For any $n \in \mathbb{N}$, we denote with $[n]$ the set $\{1, \ldots, n\}$. Due to space constraints, all proofs are deferred to the appendix.

Let $\mathcal{D}$ be a distribution defined over the *classification domain* $\mathcal{X}$. A *multiclass classification task* is specified by a *labeling function* $y : \mathcal{X} \to \mathcal{Y} = [k]$ that maps each *item* $x \in \mathcal{X}$ to a class $j$ in the label space $\mathcal{Y}$, where $k \geq 2$. We say that a multiclass classification task is *balanced* if for each $j \in [k]$, it holds that $\mathbb{P}_{x \sim \mathcal{D}}[y(x) = j] = 1/k$. Unless otherwise stated, we assume that the classification task is balanced. This assumption is not restrictive, and as we will observe later, it can be changed if a different prior is known on the class probabilities. We will show that our lower bound holds even if we do not assume balanced classes. We also assume to have access to $n$ *attribute functions* $\psi_1, \ldots, \psi_n$, where $\psi_i : \mathcal{X} \to \{0, 1\}$ for $i \in [n]$. We say that a classification item $x \in \mathcal{X}$ has attribute $i \in [n]$ if $\psi_i(x) = 1$. For ease of notation, we define $\boldsymbol{\psi}(x) \doteq (\psi_1(x), \ldots, \psi_n(x))^T$. The codomain of $\boldsymbol{\psi}$ is $\{0, 1\}^n$, and it is referred to as *attribute space*. All the information about the target unseen classes available to the algorithm is encoded in a *class-attribute matrix* $\boldsymbol{A} \in [0, 1]^{k \times n}$. The matrix provides information on the relations between classes and attributes. In particular for a class $j \in [k]$, and an attribute $i \in [n]$, $A_{j,i}$ is the probability that $\psi_i(x) = 1$ given that $y(x) = j$, i.e.,

$$A_{j,i} = \mathop{\mathbb{P}}_{x \sim \mathcal{D}}[\psi_i(x) = 1 | y(x) = j] \ . \tag{1}$$

An *attribute-class classifier* $g$ is a map from vectors in the attribute space to classes, i.e., $g : \{0, 1\}^n \to [k]$. The error of $g$ is $\varepsilon(g) \doteq \mathbb{P}_{x \sim \mathcal{D}}[g \circ \boldsymbol{\psi}(x) \neq y(x)]$. Let $\mathcal{G}$ be a collection of all the possible deterministic maps $\{0, 1\}^n \to [k]$ from the attribute space to the $k$ classes. We are interested in evaluating $\min_{g \in \mathcal{G}} \varepsilon(g)$. As we focus on the contribution of the information provided by the class-attribute matrix, we assume access to the attribute functions $\psi_1, \ldots, \psi_n$. In practice, the map to the attribute space is learned on the available labeled data for the seen classes (Lampert et al., 2014; Romera-Paredes & Torr, 2015), and it is likely noisy, and can be cause of additional error.

Let $p^*$ be the (unknown) probability mass function (PMF) of the random vector $(\psi_1(x), \ldots, \psi_n(x), y(x))$ where $x \sim \mathcal{D}$. The support of $p^*$ is $\{0, 1\}^n \times [k]$. For $\boldsymbol{v} \in \{0, 1\}^n$, and $j \in [k]$, let $p^*(\boldsymbol{v}, j) \doteq \mathbb{P}_{x \sim \mathcal{D}}[\boldsymbol{\psi}(x) = \boldsymbol{v} \wedge y(x) = j]$. The error of $g$ is a function of $p^*$:

$$\varepsilon(g) = \varepsilon(g, p^*) \doteq 1 - \sum_{\boldsymbol{v} \in \{0,1\}^n} p^*(\boldsymbol{v}, g(\boldsymbol{v})) \tag{2}$$

A function $g^* \in \mathcal{G}$ that attains minimum error $\varepsilon(g^*) = \min_{g \in \mathcal{G}} \varepsilon(g)$ is a Bayes optimal classifier with respect to $p^*$, i.e. for each $\boldsymbol{v} \in \{0, 1\}^n$, we have that $g^*(\boldsymbol{v}) = \arg \max_{j \in [k]} p^*(\boldsymbol{v}, j)$. Thus,

$$\min_{g \in \mathcal{G}} \varepsilon(g) = 1 - \sum_{\boldsymbol{v} \in \{0,1\}^n} \max_{j \in [k]} p^*(\boldsymbol{v}, j) \ . \tag{3}$$

We do not have access to labeled data for the unseen classes, so we cannot estimate $p^*$. Instead, we construct a lower bound with respect to the set of all distributions that fit the available information.

# 4 Lower Bounds for Zero-Shot Learning with Attributes

In this section, we formally define our lower bound. Consider a PMF $p$ with support over $\{0,1\}^n \times [k]$. We say that $p$ satisfies the class-attribute matrix $\boldsymbol{A}$ if (as constraints (1)) for each $i \in [n]$ and $j \in [k]$,

$$\sum_{\substack{\boldsymbol{v} \in \{0,1\}^n: \\ v_i = 1}} p(\boldsymbol{v}, j) = A_{j,i} \sum_{\boldsymbol{v} \in \{0,1\}^n} p(\boldsymbol{v}, j) \ . \tag{4}$$

Recall that $p$ is balanced if for each $j \in [k]$, it holds that $\sum_{\boldsymbol{v}} p(\boldsymbol{v}, j) = 1/k$. Let $\mathcal{P}(\boldsymbol{A})$ be the set of all possible PMFs $p$ with support over $\{0,1\}^n \times [k]$ that satisfy (4) and are balanced. Clearly, the unknown true distribution, $p^* \in \mathcal{P}(\boldsymbol{A})$. The set $\mathcal{P}(\boldsymbol{A})$ can be interpreted as the collection of all the PMFs of the random vector $(\psi_1, \ldots, \psi_n, y)$ that satisfy the constraints imposed by the information available on the prediction task and on the attribute functions. While the matrix $\boldsymbol{A}$ provides precise information on the correlation between any pair of attribute function and class, it fails to provide information on the correlation between attribute functions, i.e., it does not fully specify the distribution $p^*$. Without additional information, any PMF in $\mathcal{P}(\boldsymbol{A})$ could be equal to $p^*$. Similarly to (2), given a PMF $p \in \mathcal{P}(\boldsymbol{A})$ and an attribute-class classifier $g \in \mathcal{G}$, we can define the error of $g$ with respect to the distribution $p$ as

$$\varepsilon(g, p) \doteq 1 - \sum_{\boldsymbol{v} \in \{0,1\}^n} p(\boldsymbol{v}, g(\boldsymbol{v})) \ . \tag{5}$$

Following the computation in (3), the error of the best map from attributes to classes with respect to $p \in \mathcal{P}(\boldsymbol{A})$ is computed as

$$Q(p) \doteq \min_{g \in \mathcal{G}} \varepsilon(g, p) = 1 - \sum_{\boldsymbol{v} \in \{0,1\}^n} \max_{j \in [k]} p(\boldsymbol{v}, j) \ . \tag{6}$$

We are interested in the quantity

$$Q \doteq \max_{p \in \mathcal{P}(\boldsymbol{A})} Q(p) \tag{7}$$

i.e., $Q$ is the maximum over all distributions $p \in \mathcal{P}(\boldsymbol{A})$ of the error of the best algorithm for distribution $p$. In other words, $Q$ is the worst Bayes error with respect to all the distributions that satisfy the constraints imposed by the class-attribute matrix and on the class probabilities. Since $p^*$ can be any vector in $\mathcal{P}(\boldsymbol{A})$, the value $Q$ represents a lower bound to the best error rate that an algorithm can guarantee. In fact, without further information on the attribute functions or the prediction task, it is possible that $p^*$ attains the maximum of (7), that is in the worst-case we have that

$$\varepsilon(g, p^*) = \varepsilon(g) \geq Q \quad \forall g \in \mathcal{G}$$

In other words, the quantity $Q$ reflects a worst-case scenario where the attribute functions are correlated in such a way that it is hard to distinguish between the classes, even if the attribute functions still satisfy the constraints (1) given by the class-attribute matrix $\boldsymbol{A}$. In Section 4.3, we show that this lower bound is tight. In particular, we prove that there exists a randomized classifier from the attribute space $\{0,1\}^n$ to the classes $[k]$ whose expected error is at most $Q$ with respect to any distribution $p \in \mathcal{P}(\boldsymbol{A})$.

**Example.** Consider a balanced binary classification task with two attributes. The class-attribute matrix $\boldsymbol{A} \in \mathbb{R}^{2 \times 2}$ is such that $A_{i,j} = 1/2$ for $i, j \in \{1, 2\}$. Based on this class-attribute matrix, we consider two different scenario. In the first scenario (*best-case*), we have that items from the first class have either both attributes or none with probability $1/2$, and items from the second class have only either the first attribute or the second attribute with probability $1/2$. In this case, we can simply count the number of attributes that an item has to assign it to the correct class. In the second scenario (*worst-case*), each item has either both attributes or none with probability $1/2$ independently from the item class. In this case, any mapping from the attributes to the class is going to incur an error of $1/2$.

## 4.1 Computing the Lower Bound

In this subsection, we show how to compute $Q$ as in (7) through a Linear Program (LP). To describe a generic PMF $p$, we introduce $2^n \times k$ variables $q_{\boldsymbol{v}, j}$ with $\boldsymbol{v} \in \{0,1\}^n$ and $j \in [k]$. We use additional

$2^n$ auxiliary variables $\lambda_{\boldsymbol{v}}$, for $\boldsymbol{v} \in \{0,1\}^n$, to denote the maximums of (6), i.e. $\lambda_{\boldsymbol{v}} = \max_{j \in [k]} q_{\boldsymbol{v},j}$. The LP is formulated as follows.

$$1 - Q = \min \sum_{\boldsymbol{v}} \lambda_{\boldsymbol{v}} \tag{8}$$

$$(a) \sum_{\substack{\boldsymbol{v} \in \{0,1\}^n: \\ v_i = 1}} q_{\boldsymbol{v},j} = A_{j,i} \sum_{\boldsymbol{v} \in \{0,1\}^n} q_{\boldsymbol{v},j} \qquad \forall j \in [k], i \in [n]$$

$$(b) \sum_{\boldsymbol{v} \in \{0,1\}^n} q_{\boldsymbol{v},j} = \frac{1}{k} \qquad \forall j \in [k]$$

$$(c) \ \lambda_{\boldsymbol{v}} \geq q_{\boldsymbol{v},j} \geq 0 \qquad \forall \boldsymbol{v} \in \{0,1\}^n, j \in [k]$$

**Theorem 4.1.** *The optimal value of the LP (8) is equal to $1 - Q$, with $Q$ is as in (6).*

By removing or modifying constraint $(b)$ of the LP, it is possible to remove the assumption that the classes are balanced or provide different class weights. All the previous results still hold by changing the definition of $\mathcal{P}(\boldsymbol{A})$ accordingly. It is important to point out that since we are computing a worst-case lower bound, the class weights provide significant information. Without constraints on the class weights, the worst-case distribution could concentrate all the probability mass on few classes that are hard to differentiate using the available class-attribute matrix $\boldsymbol{A}$.

The LP has $O(k \cdot 2^n)$ variables and constraints, and therefore it is computationally expensive for large number of attributes. The dependency on $2^n$ is required to describe all the possible correlations between the output of the $n$ attribute functions. Nevertheless, we present an efficient computation for the binary case in the next subsection, and an efficient approximation for the general case in Appendix C.

### 4.2 Lower Bound for Binary Classification

In this subsection, we show how to efficiently compute $Q$ as in (7) in the case of a binary classification task, i.e. $k = 2$ and $\boldsymbol{A} = [0,1]^{2 \times n}$. For ease of notation, let

$$\boldsymbol{A} = \begin{bmatrix} \alpha_1 & \dots & \alpha_n \\ \beta_1 & \dots & \beta_n \end{bmatrix} . \tag{9}$$

**Theorem 4.2.** *Consider a balanced binary classification task and let $\boldsymbol{A}$ be as in (9). Let $Q$ be as in (7). It holds that $Q = \frac{1}{2}\left(1 - \max_{i \in [n]} |\beta_i - \alpha_i|\right)$. Moreover, let $g_a$ be the attribute-class classifier*

$$g_a(\boldsymbol{v}) = 1 + \begin{cases} v_{i^*} & \text{if } \alpha_{i^*} < \beta_{i^*} \\ 1 - v_{i^*} & \text{if } \alpha_{i^*} \geq \beta_{i^*} \end{cases}$$

*for each $\boldsymbol{v} \in \{0,1\}^n$, where $i^* = \arg\max_i |\beta_i - \alpha_i|$ and $v_i$ is the $i$-th component of the vector $\boldsymbol{v}$. Then $\varepsilon(g_a, p) = Q$ for all $p \in \mathcal{P}(\boldsymbol{A})$, i.e. the lower bound $Q$ is tight.*

The theorem shows that in the worst-case, the attributes could be correlated in such a way that it is not possible to do better than deciding solely based on the attribute with the largest gap between its probabilities in the two classes. This result also formally proves that for binary classification, the worst-case is determined by a single attribute, and there is no compounded benefit in having multiple attributes in the case of perfect attribute detectors. This result is in line with other worst-case analyses in the context of weak supervision. In Mazzetto et al. (2021b), it is noted that while combining the output of different weak supervision sources to obtain a noisy label of a given input item, in the worst-case one cannot do better than just using the most accurate weak supervision source without additional information. In Appendix C, we show how to approximate the lower bound in the multiclass setting by using Theorem 4.2.

### 4.3 Lower Bound is Tight

In this subsection, we prove that the worst-case lower bound (7) is tight. We show a randomized attribute-class classifier whose expected error is upper bounded by $Q$ with respect to any distribution $p \in \mathcal{P}(\boldsymbol{A})$. This classifier can be computed only based on the class-attribute matrix, and it provides an upper bound to the error of the best map from attributes to classes that matches the lower bound.

We consider the family $\mathcal{G}_R$ of all randomized attribute-class classifiers, where each $g \in \mathcal{G}_R$ is a random map from $\{0,1\}^n$ to $[k]$. A attribute-class classifier in $\mathcal{G}_R$ is described with a right-stochastic matrix $\boldsymbol{W} \in [0,1]^{2^n \times k}$, where the rows are indexed by binary vectors $\boldsymbol{v} \in \{0,1\}^n$, and the columns are indexed by the classes $j \in [k]$. The entry $W_{\boldsymbol{v},j}$ represents the probability of the randomized classifier to output $j$ given that the input is $\boldsymbol{v}$. We will use $g_{\boldsymbol{W}}$ to denote the randomized classifier in $\mathcal{G}_R$ that is described with the right-stochastic matrix $\boldsymbol{W}$. Given a PMF $p$ over $\{0,1\}^n \times [k]$, we define the expected error of $g_{\boldsymbol{W}}$ as

$$\varepsilon(g_{\boldsymbol{W}}, p) \doteq 1 - \mathbb{P}_{(\boldsymbol{v},j) \sim p}[g_{\boldsymbol{W}}(\boldsymbol{v}) = j] = 1 - \sum_{\boldsymbol{v} \in \{0,1\}^n} \sum_{j \in [k]} W_{\boldsymbol{v},j} \cdot p(\boldsymbol{v}, j) \ . \tag{10}$$

We can observe that the definition above extends definition (5), in fact (10) coincides with (5) if $g_{\boldsymbol{W}}$ is a deterministic classifier, i.e. each row of $\boldsymbol{W}$ contains a 1.

**Theorem 4.3.** *There exists a randomized attribute-class classifier $g_a \in \mathcal{G}_R$ such that its worst-case expected error is upper bounded by $Q$, i.e. $\max_{p \in \mathcal{P}(\boldsymbol{A})} \varepsilon(g_a, p) \leq Q$ , where $Q$ is computed as in (7). Also, it holds $\max_{p \in \mathcal{P}(\boldsymbol{A})} \min_{g \in \mathcal{G}_R} \varepsilon(g, p) = Q$, i.e. the lower bound $Q$ also applies to the family of randomized functions $\mathcal{G}_R$.*

It is possible to compute the randomized attribute-class classifier $g_a$ that satisfies Theorem 4.3 solely based on the matrix $\boldsymbol{A}$ through Linear Programming using $O(k \cdot 2^n)$ variables and constraints. Due to space constraints, we defer this computation to Appendix B.

## 5 Empirical Applications

In this section we compare our novel theory with the performance of popular attribute-based ZSL methods. Our results quantify a lower bound to the lowest error rate that any attribute-based ZSL algorithm can guarantee based on the information provided by the class-attribute matrix. In practice, we show that the lower bound is still predictive of the performance and the behaviour of attribute-based ZSL algorithms. We run two set of experiments.

1. **Comparing the lower bound and the empirical error** (Section 5.2). We compare the error rates of ZSL models with attributes with the lower bound on the error from Section 4.1.
2. **Pairwise misclassification prediction** (Section 5.3). We measure the predictive power of our lower bounds to identify pairs of classes that ZSL models are likely to misclassify. This hardness is measured using the lower bound on the error for a pair of classes (Section 4.2).

### 5.1 Experimental Setup

In this section, we briefly describe the experimental setup. Further details about the datasets and the methods can be found in Appendix D. [1] We choose the following four datasets with attributes that are widely used benchmarks in ZSL: Animals with Attributes 2 (**AwA2**) (Xian et al., 2018a), aPascal-aYahoo (**aPY**) (Farhadi et al., 2009), Caltech-UCSD Birds-200-2011 (**CUB**) (Wah et al., 2011), and SUN attribute database (**SUN**) (Patterson et al., 2014). We focus on classic ZSL algorithms with attributes that use at most the information in the class-attribute matrix for the unseen classes: **DAP** (Lampert et al., 2014), **ESZSL** (Romera-Paredes & Torr, 2015), **SAE** (Kodirov et al., 2017), **ALE** (Akata et al., 2016), **SJE** (Akata et al., 2015). We choose these methods because they use the class-attribute matrix that is the focus of our theoretical analysis. Many other ZSL methods have been proposed in recent years (see Section 2), but their comparison with our lower bound would be vacuous as they often use other source of auxiliary information on the unseen classes, and thus do not fit within our novel theoretical framework. They are beyond the scope of this first analysis of ZSL. However, we also run experiments on a more recent attribute-based method **DAZLE** (Huynh & Elhamifar, 2020) which takes advantage of additional information, i.e., attribute semantic vectors.

### 5.2 Comparing Lower Bound and Empirical Error

In this section, we compare the lower bound presented in Section 4 with the actual error of the ZSL models. To this end, we run two set of experiments: a first set using the ZSL datasets mentioned in

---
[1] Code is available at `https://github.com/BatsResearch/mazzetto-neurips22-code`.

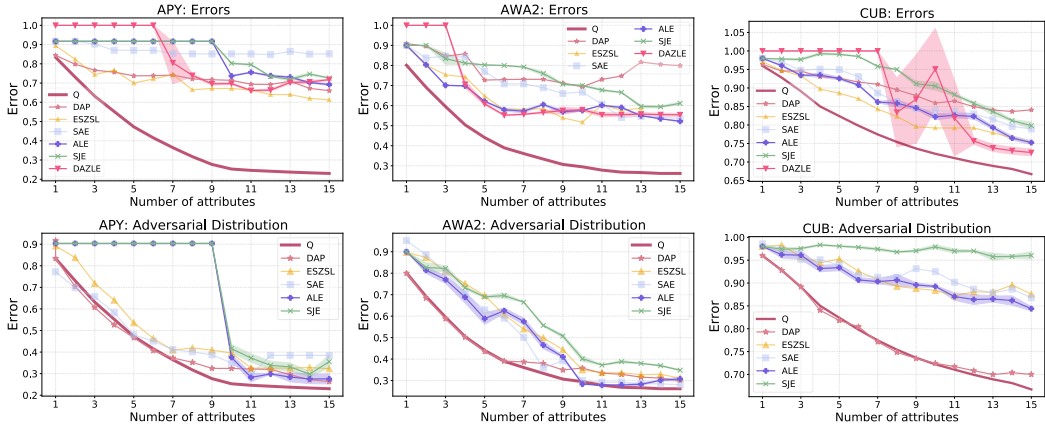

Figure 1: **Comparison of the lower bound with the empirical error.** We plot the lower bound on the error (**Q**), and the error of ZSL methods with attributes (**DAP, ESZSL, SAE, ALE**, and **DAZLE**). The first row reports these values computed on the unseen classes of the aPY, AwA2, and CUB, varying the number of available attributes. The second row reports the values for the adversarially generated synthetic data. The bands indicate the standard errors on five runs with different seeds for randomized methods. These results validate that even in the absence of domain shift, there exists a distribution of the data that satisfy the constraints imposed by the class-attribute matrix for which no method can do better than the lower bound.

the previous subsection, and a second set using adversarially generated synthetic data that conform with the class-attribute matrices of those same ZSL datasets.

In the first set of experiments, we follow the standard way to evaluate ZSL models. We train our model on the seen classes, and then compare our lower bound with the empirical error of the ZSL models on the unseen classes. Since the computation of the lower bound is very expensive for a large number of attributes (Section 4.1), we focus on a subsets of them. We propose the following greedy strategy to ensure a selection of attributes that are informative with respect to the target classes. Starting with no attributes, we iteratively add the attribute that decreases the most the value of the lower bound, up to 15 attributes. Due to the large number of seen classes of SUN and CUB, we restrict them to a smaller random subset (see Appendix D). In the first row of Figure 1, we report results for aPY, AwA2, and CUB, due to space constraints. The results for SUN are similar and in Figure 3 in Appendix E.

We observe that the value of the lower bound can be significantly lower than the error rate of the ZSL models. This gap is most probably due to the fact that the learned map from images to attributes does not generalize perfectly to the unseen classes. In fact, in this setting we can identify two main source of error for the ZSL models: (1) the arbitrary error due to the domain shift, and (2) the error due to how discriminative is the attribute space to differentiate between the different classes. Our lower bound only addresses the latter, as no method can guarantee a smaller error than the lower bound to map from attributes to classes given only the information of the class-attribute matrix. Nonetheless, for CUB and SUN we observe that the empirical error of ZSL models roughly follow the trend of the lower bound. This suggests that the lower bound can still be used as a tool to capture how the additional information provided by an attribute leads to improvements of the ZSL models.

In the second set of experiments, we empirically demonstrate our theory by showing that even if we minimize the error due to domain shift, there exists data for which no method can do better than our lower bound. To this end, for each dataset we adversarially generate synthetic data with attribute values satisfying the dataset's class-attribute matrix. Specifically, we use the same class-attribute matrix with 15 attributes as in the previous set of experiments in order to compute the adversarial distribution $p$ over attributes and classes according to the linear program introduced in Section 4.1. The data is generated by sampling attribute-class pairs from this distribution, and using the attribute vector as the feature vector. In order to minimize the error due to domain shift, this distribution is used to generate data for both training and testing of the ZSL methods, and the same class-attribute matrix is used for both seen and unseen classes. We report additional details on this experimental setup and

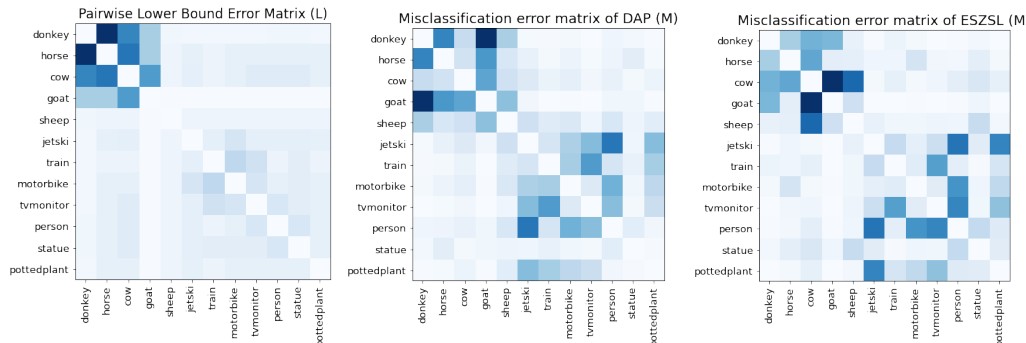

Figure 2: **Pairwise miscassification matrices.** For the unseen classes of aPY, we plot the pairwise lower bound between pair of classes $L$ (Section 4.2), and the misclassification error matrix $M$ of two ZSL models: DAP and ESZSL. Darker squares indicate higher values, and light blue on the diagonal is 0. High values of the lower bound indicate classes that are harder (in the worst-case) to distinguish in theory, and high values in $M$ indicate pair of classes that are often confused by the ZSL model.

synthetic data generation in Appendix D. We report the results of the experiments in the second row of Figure 1, iterating over the same attributes greedily selected in the first set of experiments for each dataset. (For this set of experiments, we do not report results for DAZLE as this method relies on the input items being images, so it does not apply to our synthetic data.) In this case, the methods are able to achieve errors that are comparable with the lower bound as we minimized the error due to domain shift. This experiment empirically validates that even in the absence of domain shift, there exists a distribution of the data that satisfy the constraints imposed by the class-attribute matrix for which no method can do better than the lower bound. This adversarial distribution represent an intrinsic error gap due to the quality of the information provided by the class-attribute matrix. This is the first work to quantify such information in ZSL.

## 5.3 Pairwise Misclassification Prediction

Theorem 4.2 shows how to efficiently compute the lower bound on the error to distinguish between a pair of classes given the class-attribute matrix. In addition to the overall bound on error, it also gives us fine-grained information about which classes are harder to distinguish among. We define the *pairwise lower bound error matrix* $L$, whose entry $L_{j,j'}$ is the lower bound on the error computed as in Section 4.2, for all classes $j, j' \in [k]$, $j \neq j'$. A large entry $L_{j,j'}$ between two classes $j \neq j'$ indicates that it is hard (in the worst-case) to distinguish between them. In this section, we compare the matrix $L$ with the classification errors made by the ZSL models discussed in Section 5.1. In particular, we want to show if the pairwise lower bounds on the errors are predictive of the misclassification errors made by the ZSL models. Specifically, a large lower bound on the error for a pair of classes indicates that a ZSL model would likely confuse one class with the other. For a given dataset and a ZSL method, we build a *misclassification error matrix* $M$. The entry $M_{j,j'}$ is computed as

$$\mathbb{P}_{x \sim \mathcal{D}}(h(x) = j \wedge y(x) = j' | y(x) \in \{j, j'\}) + \mathbb{P}_{x \sim \mathcal{D}}(h(x) = j' \wedge y(x) = j | y(x) \in \{j, j'\})$$

for all $j, j' \in [k]$, $j \neq j'$, where $h(\cdot)$ is the ZSL model. The entry $M_{j,j'}$ represents the probability of the model to misclassify an item of the class $j$ with the class $j'$ or vice-versa. We estimate $M$ using test data of the unseen classes.

In Figure 2, we plot $L$ together with the misclassification matrices $M$ of two ZSL methods: DAP and ESZSL, computed on the unseen classes of aPY. The pairwise lower bound matrix $L$ has large values within multiple groups of semantically similar classes, e.g., animals and transportation means. This is in line with human intuition, as we expect visually similar classes to exhibit similar attributes. Correspondingly, the misclassification matrices of DAP and ESZSL highlight the presence of many errors for classes belonging to these groups. We also note that ZSL models could misclassify other pairs of classes due to other source of errors, such as an inaccurate map from image to attributes. We report additional experimental analysis on all datasets in Appendix E.

# 6 Conclusions, Limitations, and Future Work

We present the first non-trivial lower bound on the best error that an attribute-based ZSL method can guarantee given the information provided—the class attribute matrix. While our method is limited to class-attribute matrices, it constitutes a first theoretical building block to quantify the auxiliary information provided in ZSL. In general, theoretical evaluation of the error of ZSL models remains a hard problem due to the arbitrary domain shift between seen and unseen classes, and the wide range of possible auxiliary information used. As a future direction, it remains an open problem to be able to quantify this information for other families of ZSL methods. However, our analysis readily extends to other variants of ZSL, such as generalized ZSL, where we simply use the class-attribute matrix of the union of both seen and unseen classes while computing our lower bound.

**Broader Societal Impacts**

Zero-shot learning is now a popular scenario in research, with potential application to real-world language and vision tasks. Worse-case guarantees have long been desired in ZSL. Any improvement in the rigor of claims about model performance has impact because it demonstrates both what performance can be achieved and that some solutions are invalid. However, such bounds do not cover many kinds of error, such as a generalization gap from domain shift or label errors. Further, it is important that bounds are correctly interpreted such that no false claims or confidences are drawn from our findings. An educated interpretation of the effect of these bounds upon any particular machine learning application is still required.

**Acknowledgements**

We thank Michael Littman and James Tompkin for many helpful and insightful discussions. This material is based on research sponsored by Defense Advanced Research Projects Agency (DARPA) and Air Force Research Laboratory (AFRL) under agreement number FA8750-19-2-1006 and by the National Science Foundation (NSF) under award IIS-1813444. The U.S. Government is authorized to reproduce and distribute reprints for Governmental purposes notwithstanding any copyright notation thereon. The views and conclusions contained herein are those of the authors and should not be interpreted as necessarily representing the official policies or endorsements, either expressed or implied, of Defense Advanced Research Projects Agency (DARPA) and Air Force Research Laboratory (AFRL) or the U.S. Government. We gratefully acknowledge support from Google and Cisco. Disclosure: Stephen Bach is an advisor to Snorkel AI, a company that provides software and services for weakly supervised machine learning.

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
