# A Deferred Proofs.

**Proof of Theorem 4.1.** Let $Q'$ be the optimal value of the LP.

Let $p^* \in \mathcal{P}(\boldsymbol{A})$ be a solution of the maximization (7). Consider the following assignment of the variables $q_{\boldsymbol{v},j} = p^*(\boldsymbol{v}, j)$ for all $\boldsymbol{v} \in \{0,1\}^n$ and $j \in [k]$. Since $p^* \in \mathcal{P}(\boldsymbol{A})$, it is straight-forward to verify that the variables $q_{\boldsymbol{v},j}$ satisfy constraints $(a)$ and $(b)$ of the LP. Moreover, the objective function is minimized whenever the values $\lambda_{\boldsymbol{v}}$ are chosen as small as possible. Due to constraint $(c)$ of the LP, we have that $\lambda_{\boldsymbol{v}} = \max_{j \in [k]} q_{\boldsymbol{v},j}$ for each $\boldsymbol{v} \in \{0,1\}^n$. We have that

$$1 - Q = \sum_{\boldsymbol{v} \in \{0,1\}^n} \max_{\boldsymbol{v} \in \{0,1\}^n} \max_{j \in [k]} p^*(\boldsymbol{v}, j) = \sum_{\boldsymbol{v} \in \{0,1\}^n} \lambda_{\boldsymbol{v}} \geq 1 - Q' \tag{11}$$

By contradiction, assume the optimal solution $q_{\boldsymbol{v},j}^*, \lambda_{\boldsymbol{v}}^*$ is such that $1 - Q' = \sum_{\boldsymbol{v} \in \{0,1\}^n} \lambda_{\boldsymbol{v}}^* < 1 - Q$. Since $q_{\boldsymbol{v},j}^*, \lambda_{\boldsymbol{v}}^*$ is an optimal solution, due to constraint $(c)$ we have that $\lambda_{\boldsymbol{v}}^* = \max_{j \in [k]} q_{\boldsymbol{v},j}^*$. Consider a PMF $\tilde{p}$ over $\{0,1\}^n \times [k]$ such that $\tilde{p}(\boldsymbol{v}, j) = q_{\boldsymbol{v},j}^*$. It is easy to verify that $\tilde{p} \in \mathcal{P}(\boldsymbol{A})$ due to the constraint $(a)$ and $(b)$. Moreover, we have that $Q(\tilde{p}) = 1 - \sum_{\boldsymbol{v}} \max \tilde{p}(\boldsymbol{v}, j) = 1 - \sum_{\boldsymbol{v}} \lambda_{\boldsymbol{v}}^* > Q$. This is a contradiction as $\max_{p \in \mathcal{P}(\boldsymbol{A})} Q(p) = Q$. Therefore, we have that $1 - Q' \geq 1 - Q$. Combining the latter inequality with inequality (11), we can conclude that $Q = Q'$.

$\square$

**Proof of Theorem 4.2.** Without loss of generality, we assume that $\alpha_i \geq \beta_i$ for each $i \in [n]$. In fact, if $\alpha_i < \beta_i$, then we can consider the attribute function $\psi_i' = 1 - \psi_i$, and the $i$-th column of the matrix $\boldsymbol{A}$ would become $(1 - \alpha_i, 1 - \beta_i)^T$, with $1 - \alpha_i \geq 1 - \beta_i$. Also, assume that the attributes are ordered such that $\alpha_1 - \beta_1 \geq \alpha_i - \beta_i$ for each $i \in [n]$.

We first prove the second part of the Theorem. Let $g_a$ be defined as in the problem statement. It is easy to see that for any $p \in \mathcal{P}(\boldsymbol{A})$, we have that

$$\varepsilon(g_a, p) = \mathop{\mathbb{P}}_{x \sim \mathcal{D}} (g_a \circ \boldsymbol{\psi}(x) \neq y(x)) = \tag{12}$$
$$= \mathbb{P}(\psi_1(x) = 0 | y(x) = 1) \mathbb{P}(y(x) = 1) + \mathbb{P}(\psi_1(x) = 1 | y(x) = 0) \mathbb{P}(y(x) = 0)$$
$$= \frac{1}{2}(1 - \alpha_1) + \frac{1}{2}\beta_1$$
$$= \frac{1}{2}(1 - |\beta_1 - \alpha_1|) = Q$$

Since this holds for any $p \in \mathcal{P}(\boldsymbol{A})$, we have that

$$\max_{p \in \mathcal{P}(\boldsymbol{A})} \varepsilon(g_a, p) = \frac{1}{2}(1 - |\beta_1 - \alpha_1|) \ .$$

Now, we will prove the first part of the Theorem. The proof is by induction. For $i \in [n]$, let $\boldsymbol{A}_i$ be the matrix that consists of the first $i$ columns of $\boldsymbol{A}$. For $i \in [n]$, let $\mathcal{G}_i$ be the set of all the functions $\{0,1\}^n \to [2]$. For $i \in [n]$, let $p^{(i)}$ be a PMF with support over $\{0,1\}^i \times [2]$ such that

$$p^{(i)} = \arg\max_{p \in \mathcal{P}(\boldsymbol{A}_i)} \min_{g \in \mathcal{G}_i} \underbrace{\left(1 - \sum_{\boldsymbol{v} \in \{0,1\}^i} p(\boldsymbol{v}, g(\boldsymbol{v}))\right)}_{\stackrel{\doteq}{=} \varepsilon^{(i)}(g,p)} \tag{13}$$

For ease of notation, for $i \in [n]$, we will denote $p_{\boldsymbol{v},j}^{(i)} = p^{(i)}(\boldsymbol{v}, j)$ for each $\boldsymbol{v} \in \{0,1\}^i$ and $j \in [2]$.

The following auxiliary proposition is crucial to prove the theorem.

**Proposition A.1.** *Let $i \in [n]$. We have that $\min_{g \in \mathcal{G}_i} \varepsilon^{(i)}(g, p^{(i)}) = Q$ if and only if for each $\boldsymbol{v} \in \{0,1\}^{i-1}$, it holds both $p_{1\boldsymbol{v},1}^{(i)} \geq p_{1\boldsymbol{v},2}^{(i)}$ and $p_{0\boldsymbol{v},1}^{(i)} \leq p_{0\boldsymbol{v},2}^{(i)}$.*

*Proof.* Assume that for each $\boldsymbol{v} \in \{0,1\}^{i-1}$, it holds both $p_{1\boldsymbol{v},1}^{(i)} \geq p_{1\boldsymbol{v},2}^{(i)}$ and $p_{0\boldsymbol{v},1}^{(i)} \leq p_{0\boldsymbol{v},2}^{(i)}$. Then, we have that

$$\min_{g \in \mathcal{G}_i} \varepsilon^{(i)}(g, p^{(i)}) = 1 - \sum_{\boldsymbol{v} \in \{0,1\}^i} \max(p_{\boldsymbol{v},1}^{(i)}, p_{\boldsymbol{v},2}^{(i)}) =$$

$$= 1 - \sum_{\boldsymbol{v} \in \{0,1\}^{i-1}} \max(p_{0\boldsymbol{v},1}^{(i)}, p_{0\boldsymbol{v},2}^{(i)}) - \sum_{\boldsymbol{v} \in \{0,1\}^{i-1}} \max(p_{1\boldsymbol{v},1}^{(i)}, p_{1\boldsymbol{v},2}^{(i)})$$

$$= 1 - \sum_{\boldsymbol{v} \in \{0,1\}^{i-1}} p_{0\boldsymbol{v},2}^{(i)} - \sum_{\boldsymbol{v} \in \{0,1\}^{i-1}} p_{1\boldsymbol{v},1}^{(i)}$$

$$= 1 - \frac{1}{2}(1 - \beta_1) + \frac{1}{2}\alpha_1 \tag{14}$$

$$= \frac{1}{2}\left(1 - |\alpha_1 - \beta_1|\right) = Q$$

Equality (14) is simply obtained by marginalization, since $p^{(i)} \in \mathcal{P}(\boldsymbol{A}_i)$, thus $\mathbb{P}_{x \sim \mathcal{D}}(\psi_1(x) = 0 \wedge y(x) = 2) = (1 - \beta_1)/2$ and $\mathbb{P}_{x \sim \mathcal{D}}(\psi_1(x) = 1 \wedge y(x) = 1) = \alpha_1/2$.

Assume that there exists $\boldsymbol{v}' \in \{0,1\}^{i-1}$ such that $p_{1\boldsymbol{v}',1}^{(i)} < p_{1\boldsymbol{v}',2}^{(i)}$ (the case $p_{0\boldsymbol{v}',1}^{(i)} > p_{0\boldsymbol{v}',2}^{(i)}$ is proven with the same argument). Let $g_a^{(i)}$ be defined similarly to $g_a$, i.e. $g_a^{(i)} = 1$ if $\psi_1(x) = 1$, and $g_a^{(i)} = 2$ otherwise. Following the same computation of (12), we can show that $\varepsilon^{(i)}(g_a, p^{(i)}) = Q$. Consider the classifier $\tilde{g}$ such that $\tilde{g}(\boldsymbol{v}) = g_a(\boldsymbol{v})$ for all $\boldsymbol{v} \in \{0,1\}^i$ such that $\boldsymbol{v} \neq 1\boldsymbol{v}'$, and $\tilde{g}(1\boldsymbol{v}') = 2$. We have that

$$\varepsilon^{(i)}(g_a^{(i)}, p^{(i)}) - \varepsilon^{(i)}(\tilde{g}, p^{(i)}) = p^{(i)}(\boldsymbol{v}', \tilde{g}(\boldsymbol{v}')) - p^{(i)}(\boldsymbol{v}', g_a^{(i)}(\boldsymbol{v})) = p_{1\boldsymbol{v}',2}^{(i)} - p_{1\boldsymbol{v}',1}^{(i)} > 0$$

Therefore, $\varepsilon^{(i)}(\tilde{g}, p^{(i)}) < \varepsilon^{(i)}(g_a^{(i)}, p^{(i)}) = Q$, which directly implies that $\min_{g \in \mathcal{G}_i} \varepsilon^{(i)}(g, p^{(i)}) < Q$. □

By induction, we will prove that for each $i \in [n]$, it is true that $\min_{g \in \mathcal{G}} \varepsilon^{(i)}(g, p^{(i)}) = Q$.

**Base case.** Let $i = 1$. We have that

$$p_{1,1}^{(1)} = \frac{\alpha_1}{2} \qquad\qquad p_{0,1}^{(1)} = \frac{1}{2}(1 - \alpha_1)$$

$$p_{1,2}^{(1)} = \frac{\beta_1}{2} \qquad\qquad p_{0,2}^{(1)} = \frac{1}{2}(1 - \beta_1)$$

Observe that $p^{(1)} \in \mathcal{P}(\boldsymbol{A}_1)$ as the classes are balanced, and we satisfy the constraints of the matrix $\boldsymbol{A}$ for the first attribute. It is easy to observe that

$$\min_{g \in \mathcal{G}} \varepsilon^{(1)}(g, p^{(1)}) = 1 - \frac{\alpha_1}{2} - \frac{1}{2}(1 - \beta_1) = Q$$

**Inductive step.** For $i \in 2, \ldots, n$, assume that $\min_{g \in \mathcal{G}_{i-1}} \varepsilon^{(i-1)}(g, p^{(i-1)}) = Q$, where $p^{(i-1)}$ is solution of (13). We will show how to construct $p^{(i)}$ from $p^{(i-1)}$ guaranteeing $\min_{g \in \mathcal{G}_i} \varepsilon^{(i)}(g, p^{(i)}) = Q$ and that $p^{(i)} \in \mathcal{P}(\boldsymbol{A}_i)$. Observe that in that case, $p^{(i)}$ is also a solution of (13), since the classifier $g_a^{(i)}$ (defined as in the proof of Proposition A.1) has error exactly $Q$ with respect to any PMF $p \in \mathcal{P}(\boldsymbol{A}_i)$.

Our construction will be divided in three different cases, based on the ordering of the values $\alpha_1, \beta_1$, $\alpha_i$ and $\beta_i$. We will exhibit a different construction of $p^{(i)}$ for each of the case, but they all share the same proof line. In particular, we will guarantee that for each $\boldsymbol{v} \in \{0,1\}^{i-1}$ and $j \in [2]$, it holds

$$p_{\boldsymbol{v}1,j}^{(i)} + p_{\boldsymbol{v}0,j}^{(i)} = p_{\boldsymbol{v},j}^{(i-1)} \tag{15}$$

This immediately implies that the classes are balanced, in fact, for any $j \in [2]$, we have that

$$\sum_{\boldsymbol{v} \in \{0,1\}^i} p_{\boldsymbol{v},j}^{(i)} = \sum_{\boldsymbol{v} \in \{0,1\}^{i-1}} p_{\boldsymbol{v}0,j}^{(i)} + p_{\boldsymbol{v}1,j}^{(i)} = \sum_{\boldsymbol{v} \in \{0,1\}^{i-1}} p_{\boldsymbol{v},j}^{(i-1)} = \frac{1}{2} ,$$

where the last inequality is due to the assumption that $p^{(i-1)} \in \mathcal{P}(\boldsymbol{A}_{i-1})$. Moreover, (15) also implies that $p^{(i)}$ satisfies the constraints imposed by the matrix $\boldsymbol{A}$ for the first $i-1$ attributes. In fact, for any $a \in [i-1]$, and $j \in [2]$, we have that

$$
\sum_{\boldsymbol{v} \in \{0,1\}^i : v_a = 1} p^{(i)}_{\boldsymbol{v},j} = A_{j,a} \sum_{\boldsymbol{v} \in \{0,1\}^i} p^{(i)}_{\boldsymbol{v},j}
$$

$$
\iff \sum_{\boldsymbol{v} \in \{0,1\}^{i-1} : v_a = 1} \left( p^{(i)}_{\boldsymbol{v}0,j} + p^{(i)}_{\boldsymbol{v}1,j} \right) = A_{j,a} \sum_{\boldsymbol{v} \in \{0,1\}^{i-1}} \left( p^{(i)}_{\boldsymbol{v}0,j} + p^{(i)}_{\boldsymbol{v}1,j} \right)
$$

$$
\iff \sum_{\boldsymbol{v} \in \{0,1\}^{i-1} : v_a = 1} p^{(i-1)}_{\boldsymbol{v},j} = A_{j,a} \sum_{\boldsymbol{v} \in \{0,1\}^{i-1}} p^{(i-1)}_{\boldsymbol{v},j} \ .
$$

The latter equality is true as $p^{(i-1)} \in \mathcal{P}(\boldsymbol{A}_{i-1})$.

For each different case, we will show that our construction also satisfies the constraints imposed by matrix $\boldsymbol{A}$ for attribute $i$. This, together with (15), implies that our construction guarantees that $p^{(i)} \in \mathcal{P}(\boldsymbol{A})$.

Moreover, we will show that with our construction, we also guarantee that for each $\boldsymbol{v} \in \{0,1\}^{i-1}$, it holds that

$$
p^{(i)}_{1\boldsymbol{v},1} \geq p^{(i)}_{1\boldsymbol{v},2} \ \wedge \ p^{(i)}_{0\boldsymbol{v},1} \leq p^{(i)}_{0\boldsymbol{v},2} \ . \tag{16}
$$

Using Proposition A.1, (16) immediately implies that $\min_{g \in \mathcal{G}_i} \varepsilon^{(i)}(g, p^{(i)}) = Q$. In order to show that (16) holds in our construction, we will use the fact that for each $\boldsymbol{v} \in \{0,1\}^{i-2}$, it holds that

$$
p^{(i-1)}_{1\boldsymbol{v},1} \geq p^{(i-1)}_{1\boldsymbol{v},2} \ \wedge \ p^{(i-1)}_{0\boldsymbol{v},1} \leq p^{(i-1)}_{0\boldsymbol{v},2} \ . \tag{17}
$$

This is indeed the case, as by assumption $\min_{g \in \mathcal{G}_{i-1}} \varepsilon^{(i-1)}(g, p^{(i-1)}) = Q$, hence we can apply the other direction of Proposition A.1.

We will now show our construction for the three different cases. For each case, it is straightforward to check that in our construction (15) holds, and that (17) immediately implies (16). Therefore, we omit those computations.

**First Case**. $[\beta_1 \geq \beta_i \wedge \alpha_i \leq \alpha_1]$. We construct $p^{(i)}$ as follows. For each $\boldsymbol{v} \in \{0,1\}^{i-2}$, we let

$$
p^{(i)}_{1\boldsymbol{v}1,2} = p^{(i-1)}_{1\boldsymbol{v},2} \qquad\qquad\qquad p^{(i)}_{1\boldsymbol{v}0,2} = 0
$$

$$
p^{(i)}_{1\boldsymbol{v}1,1} = p^{(i-1)}_{1\boldsymbol{v},2} + \frac{\alpha_i - \beta_1}{\alpha_1 - \beta_1} \left( p^{(i-1)}_{1\boldsymbol{v},1} - p^{(i-1)}_{1\boldsymbol{v},2} \right)
$$

$$
p^{(i)}_{1\boldsymbol{v}0,1} = \frac{\alpha_1 - \alpha_i}{\alpha_1 - \beta_1} \left( p^{(i-1)}_{1\boldsymbol{v},1} - p^{(i-1)}_{1\boldsymbol{v},2} \right) \ .
$$

These probabilities are well defined, as $0 \leq \frac{\alpha_i - \beta_1}{\alpha_1 - \beta_1} \leq 1$ and $p^{(i-1)}_{1\boldsymbol{v},1} \geq p^{(i-1)}_{1\boldsymbol{v},2}$. By construction, we have that $p^{(i)}_{1\boldsymbol{v}1,2} + p^{(i)}_{1\boldsymbol{v}0,2} = p^{(i-1)}_{1\boldsymbol{v},2}$ and $p^{(i)}_{1\boldsymbol{v}1,1} + p^{(i)}_{1\boldsymbol{v}0,1} = p^{(i-1)}_{1\boldsymbol{v},1}$, and it is easy to see that $p^{(i)}_{1\boldsymbol{v}1,1} \geq p^{(i)}_{1\boldsymbol{v}1,2}$ and $p^{(i)}_{1\boldsymbol{v}0,1} \geq p^{(i)}_{1\boldsymbol{v}0,2}$.

For each $\boldsymbol{v} \in \{0,1\}^{i-2}$, we let

$$
p^{(i)}_{0\boldsymbol{v}0,1} = p^{(i-1)}_{0\boldsymbol{v},1} \qquad\qquad\qquad p^{(i)}_{0\boldsymbol{v}1,1} = 0
$$

$$
p^{(i)}_{0\boldsymbol{v}0,2} = p^{(i-1)}_{0\boldsymbol{v},1} + \frac{\alpha_1 - \beta_i}{\alpha_1 - \beta_1} \left( p^{(i-1)}_{0\boldsymbol{v},2} - p^{(i-1)}_{0\boldsymbol{v},1} \right)
$$

$$
p^{(i)}_{0\boldsymbol{v}1,2} = \frac{\beta_i - \beta_1}{\alpha_1 - \beta_1} \left( p^{(i-1)}_{0\boldsymbol{v},2} - p^{(i-1)}_{0\boldsymbol{v},1} \right)
$$

Again, by construction we have that $p^{(i)}_{0\boldsymbol{v}0,2} + p^{(i)}_{0\boldsymbol{v}1,2} = p^{(i-1)}_{0\boldsymbol{v},2}$ and $p^{(i)}_{0\boldsymbol{v}0,1} + p^{(i)}_{0\boldsymbol{v}1,1} = p^{(i-1)}_{0\boldsymbol{v},1}$, and it is easy to see that $p^{(i)}_{0\boldsymbol{v}0,2} \geq p^{(i)}_{0\boldsymbol{v}0,1}$ and $p^{(i)}_{0\boldsymbol{v}1,2} \geq p^{(i)}_{0\boldsymbol{v}1,1}$.

The PMF $p^{(i)}$ satisfies the constraints imposed by the class-attribute matrix $\boldsymbol{A}$ for the attribute $i$, in fact

$$\sum_{\boldsymbol{v}\in\{0,1\}^{i-1}} p^{(i)}_{\boldsymbol{v}1,1} = \frac{\beta_1}{2} + \frac{\alpha_i - \beta_1}{\alpha_1 - \beta_1} \cdot \frac{1}{2}(\alpha_1 - \beta_1) = \frac{\alpha_i}{2}$$

$$\sum_{\boldsymbol{v}\in\{0,1\}^{i-1}} p^{(i)}_{\boldsymbol{v}1,2} = \frac{\beta_1}{2} + \frac{\beta_i - \beta_1}{\alpha_1 - \beta_1} \cdot \frac{1}{2}(\alpha_1 - \beta_1) = \frac{\beta_i}{2}$$

**Second case.** $[\beta_1 \leq \beta_i \wedge \alpha_1 \leq \alpha_i]$. We construct $p^{(i)}$ as follows. For each $\boldsymbol{v} \in \{0,1\}^{i-2}$, let

$$p^{(i)}_{1\boldsymbol{v}1,1} = p^{(i-1)}_{1\boldsymbol{v},1} \qquad\qquad p^{(i)}_{1\boldsymbol{v}0,1} = 0$$
$$p^{(i)}_{1\boldsymbol{v}1,2} = p^{(i-1)}_{1\boldsymbol{v},2} \qquad\qquad p^{(i)}_{1\boldsymbol{v}0,2} = 0$$

and let

$$p^{(i)}_{0\boldsymbol{v}1,1} = \frac{\alpha_i - \alpha_1}{1 - \alpha_1} p^{(i-1)}_{0\boldsymbol{v},1}$$

$$p^{(i)}_{0\boldsymbol{v}0,1} = \frac{1 - \alpha_1}{1 - \alpha_1} p^{(i-1)}_{0\boldsymbol{v},1}$$

$$p^{(i)}_{0\boldsymbol{v}1,2} = \frac{\alpha_i - \alpha_1}{1 - \alpha_1} p^{(i-1)}_{0\boldsymbol{v},1} + \frac{(\alpha_1 - \beta_1) - (\alpha_i - \beta_i)}{\alpha_1 - \beta_1} \left( p^{(i-1)}_{0\boldsymbol{v},2} - p^{(i-1)}_{0\boldsymbol{v},1} \right)$$

$$p^{(i)}_{0\boldsymbol{v}0,2} = \frac{1 - \alpha_1}{1 - \alpha_1} p^{(i-1)}_{0\boldsymbol{v},1} + \frac{\alpha_i - \beta_i}{\alpha_1 - \beta_1} \left( p^{(i-1)}_{0\boldsymbol{v},2} - p^{(i-1)}_{0\boldsymbol{v},1} \right)$$

By construction, we can observe that for each $\boldsymbol{v} \in \{0,1\}^{i-1}$, it holds that $p^{(i)}_{1\boldsymbol{v},1} \geq p^{(i)}_{1\boldsymbol{v},2}$ and that $p^{(i)}_{0\boldsymbol{v},2} \geq p^{(i)}_{0\boldsymbol{v},1}$. Moreover, for each $\boldsymbol{v} \in \{0,1\}^i$ and $j \in [2]$, it holds that $p^{(i)}_{\boldsymbol{v}1,j} + p^{(i)}_{\boldsymbol{v}0,j} = p^{(i-1)}_{\boldsymbol{v},j}$.

The PMF $p^{(i)}$ satisfies the constraints imposed by the class-attribute matrix $\boldsymbol{A}$ for the attribute $i$, in fact

$$\sum_{\boldsymbol{v}\in\{0,1\}^{i-1}} p^{(i)}_{\boldsymbol{v}1,1} = \frac{\alpha_1}{2} + \frac{\alpha_i - \alpha_1}{1 - \alpha_1} \cdot \frac{1}{2}(1 - \alpha_1) = \frac{\alpha_i}{2}$$

$$\sum_{\boldsymbol{v}\in\{0,1\}^{i-1}} p^{(i)}_{\boldsymbol{v}1,2} = \frac{\beta_1}{2} + \frac{\alpha_i - \alpha_1}{1 - \alpha_1} \cdot \frac{1}{2}(1 - \alpha_1) +$$

$$+ \frac{(\alpha_1 - \beta_1) - (\alpha_i - \beta_i)}{\alpha_1 - \beta_1} \cdot \frac{1}{2}(\alpha_1 - \beta_1) = \frac{\beta_i}{2}$$

**Third case.** $[\beta_i \leq \beta_1 \wedge \alpha_i \leq \alpha_1]$. We construct $p^{(i)}$ as follows. For each $\boldsymbol{v} \in \{0,1\}^{i-2}$, let

$$p^{(i)}_{0\boldsymbol{v}0,1} = p^{(i-1)}_{0\boldsymbol{v},1} \qquad\qquad p^{(i)}_{0\boldsymbol{v}1,1} = 0$$
$$p^{(i)}_{0\boldsymbol{v}0,2} = p^{(i-1)}_{0\boldsymbol{v},2} \qquad\qquad p^{(i)}_{0\boldsymbol{v}1,2} = 0$$

and let

$$p^{(i)}_{1\boldsymbol{v}1,2} = \frac{\beta_i}{\beta_1} p^{(i-1)}_{1\boldsymbol{v},2}$$

$$p^{(i)}_{1\boldsymbol{v}0,2} = \frac{\beta_1 - \beta_i}{\beta_1} p^{(i-1)}_{1\boldsymbol{v},2}$$

$$p^{(i)}_{1\boldsymbol{v}1,1} = \frac{\beta_i}{\beta_1} p^{(i-1)}_{1\boldsymbol{v},2} + \frac{\alpha_i - \beta_i}{\alpha_1 - \beta_1} \left( p^{(i-1)}_{1\boldsymbol{v},1} - p^{(i-1)}_{1\boldsymbol{v},2} \right)$$

$$p^{(i)}_{1\boldsymbol{v}0,1} = \frac{\beta_1 - \beta_i}{\beta_1} p^{(i-1)}_{1\boldsymbol{v},2} + \frac{(\alpha_1 - \beta_1) - (\alpha_i - \beta_i)}{\alpha_1 - \beta_1} \left( p^{(i-1)}_{1\boldsymbol{v},1} - p^{(i-1)}_{1\boldsymbol{v},2} \right)$$

Again, by construction, we can observe that for each $\boldsymbol{v} \in \{0,1\}^{i-1}$, it holds that $p^{(i)}_{1\boldsymbol{v},1} \geq p^{(i)}_{1\boldsymbol{v},2}$ and that $p^{(i)}_{0\boldsymbol{v},2} \geq p^{(i)}_{0\boldsymbol{v},1}$. Moreover, for each $\boldsymbol{v} \in \{0,1\}^i$ and $j \in [2]$, it holds that $p^{(i)}_{\boldsymbol{v}1,j} + p^{(i)}_{\boldsymbol{v}0,j} = p^{(i-1)}_{\boldsymbol{v},j}$.

The PMF $p^{(i)}$ satisfies the constraints imposed by the class-attribute matrix $\boldsymbol{A}$ for the attribute $i$, in fact

$$\sum_{\boldsymbol{v}\in\{0,1\}^{i-1}} p_{\boldsymbol{v}1,1}^{(i)} = \frac{\beta_i}{\beta_1}\frac{\beta_1}{2} + \frac{\alpha_i - \beta_i}{\alpha_1 - \beta_1}\cdot\frac{1}{2}(\alpha_1 - \beta_1) = \frac{\alpha_i}{2}$$

$$\sum_{\boldsymbol{v}\in\{0,1\}^{i-1}} p_{\boldsymbol{v}1,2}^{(i)} = \frac{\beta_i}{\beta_1}\cdot\frac{1}{2}\beta_1 = \frac{\beta_i}{2}$$

We conclude the proof by observing that since $\alpha_1 - \beta_1 \geq \alpha_i - \beta_i$, the case $[\beta_i < \beta_1 \wedge \alpha_1 < \alpha_i]$ is impossible. $\qquad\square$

**Proof of Theorem 4.3.**
By combining (6) and (7), we can rewrite $Q$ as

$$Q = \max_{p\in\mathcal{P}(\boldsymbol{A})} \min_{g\in\mathcal{G}} \varepsilon(g,p) \ .$$

Consider the maximin

$$Q' = \max_{p\in\mathcal{P}(\boldsymbol{A})} \min_{g_{\boldsymbol{W}}\in\mathcal{G}_R} \varepsilon(g_{\boldsymbol{W}},p) \ . \tag{18}$$

We show that $Q = Q'$. In fact, given $p \in \mathcal{P}(\boldsymbol{A})$, it is clear that the expected error (10) of a randomized attribute-class classifier $g_{\boldsymbol{W}} \in \mathcal{G}_R$

$$\varepsilon(g_{\boldsymbol{W}},p) = 1 - \sum_{\boldsymbol{v}\in\{0,1\}^n} \sum_{j\in[k]} W_{\boldsymbol{v},j}\cdot p(\boldsymbol{v},j)$$

is minimized when $\boldsymbol{W}_{\boldsymbol{v},j'} = 1$ if $j' = \arg\max_{j\in[k]} p(\boldsymbol{v},j)$ for all $\boldsymbol{v} \in \{0,1\}^n$, and such a attribute-class classifier is deterministic, i.e. it is equal with probability 1 to a properly chosen classifier in $\mathcal{G}$. This proves the second part of the Theorem.

Given $\alpha \in [0,1]$ and $p_1, p_2 \in \mathcal{P}(\boldsymbol{A})$, we define $p_\alpha = \alpha p_1 + (1-\alpha)p_2$ as a convex combination of $p_1$ and $p_2$, where for each $\boldsymbol{v} \in \{0,1\}^n$ and $j \in [k]$, we have that $p_\alpha(\boldsymbol{v},j) = \alpha p_1(\boldsymbol{v},j) + (1-\alpha)p_2(\boldsymbol{v},j)$. It is easy to verify that $p_\alpha \in \mathcal{P}(\boldsymbol{A})$. Moreover, for two randomized attribute-class classifiers $g_{\boldsymbol{W}}, g_{\boldsymbol{W}'}$, and $\alpha \in [0,1]$ we define $g_\alpha = g_{\alpha\boldsymbol{W}+(1-\alpha)\boldsymbol{W}'}$ as the convex combination of $g_{\boldsymbol{W}}$ and $g_{\boldsymbol{W}'}$, and observe that $g_\alpha \in \mathcal{G}_R$.

The sets $\mathcal{P}(\boldsymbol{A})$ and $\mathcal{W}$ are closed and bounded, therefore compact, and we have shown they are also convex. Moreover, the function $\epsilon(\cdot,\cdot)$ is bilinear with respect to $p$ and $\boldsymbol{W}$. Therefore, by von Neumann's Minimax Theorem (Neumann, 1928), the value of the minimax is equal to the value of the maximin, i.e.

$$\min_{g\in\mathcal{G}_R} \max_{p\in\mathcal{P}(\boldsymbol{A})} \varepsilon(g_{\boldsymbol{W}},p) = \max_{p\in\mathcal{P}(\boldsymbol{A})} \min_{g\in\mathcal{G}_R} \varepsilon(g_{\boldsymbol{W}},p) = Q$$

$$\qquad\square$$

# B  Adversarial Attribute-Class Classifier Computation

In this section of the Appendix, we show how to compute a randomized attribute-class classifier that satisfies Theorem 4.3. First, we show that the randomized attribute-class classifier

$$g_{\boldsymbol{W}^*} = \arg\min_{g_{\boldsymbol{W}}\in\mathcal{G}_R} \max_{p\in\mathcal{P}(\boldsymbol{A})} \varepsilon(g_{\boldsymbol{W}},p) \tag{19}$$

satisfies the condition of the Theorem. In fact, as noted in the proof of Theorem 4.3, we have that

$$\min_{g\in\mathcal{G}_R} \max_{p\in\mathcal{P}(\boldsymbol{A})} \varepsilon(g_{\boldsymbol{W}},p) = \max_{p\in\mathcal{P}(\boldsymbol{A})} \min_{g\in\mathcal{G}_R} \varepsilon(g_{\boldsymbol{W}},p) = \max_{p\in\mathcal{P}(\boldsymbol{A})} \min_{g\in\mathcal{G}} \varepsilon(g,p) = Q$$

Now, we show how to compute $\boldsymbol{W}^*$. The minimax (19) can be written as a bilinear problem. Let $w_{\boldsymbol{v},j}$ and $q_{\boldsymbol{v},j}$ be variables that denote respectively $W_{\boldsymbol{v},j}$ and $p(\boldsymbol{v},j)$ for $\boldsymbol{v} \in \{0,1\}^n$ and $j \in [k]$.

Inspecting (10), and using the fact that the minimax is equal to the maximin, we can compute (19) as

$$1 + \max_{\boldsymbol{q} \geq 0} \min_{\boldsymbol{w} \geq 0} \sum_{\boldsymbol{v} \in \{0,1\}^n} \sum_{j \in [k]} (-w_{\boldsymbol{v},j}) \cdot q_{\boldsymbol{v},j} \qquad\qquad s.t. \qquad (20)$$

$$(a) \sum_{\substack{\boldsymbol{v} \in \{0,1\}^n: \\ v_i = 1}} q_{\boldsymbol{v},j} = A_{j,i} P_j \qquad\qquad \forall j \in [k], i \in [n]$$

$$(b) \sum_{\boldsymbol{v} \in \{0,1\}^n} q_{\boldsymbol{v},j} = P_j \qquad\qquad \forall j \in [k]$$

$$(c) \sum_{j \in [k]} w_{\boldsymbol{v},j} = 1 \qquad\qquad \forall \boldsymbol{v} \in \{0,1\}^n$$

We use $P_j$ to denote $P_j = \mathbb{P}_{x \sim \mathcal{D}}(y(x) = j)$ for $j \in [k]$, which is equal to $1/k$ when the classes are balanced. We transform the maximin (20) in a minimization problem that can be easily solved using a Linear Program.

For a given $\boldsymbol{q}$, let $\boldsymbol{w}^{\boldsymbol{q}}$ be an assignment of the variables $\boldsymbol{w}$ that achieves the minimum. We can write the dual of the maximization problem over the variables $\boldsymbol{q}$ with respect to the fixed $\boldsymbol{w}^{\boldsymbol{q}}$ as

$$1 + \min_{\substack{\boldsymbol{a} \in \mathbb{R}^k \\ \boldsymbol{b} \in \mathbb{R}^{k \times n}}} \left( \sum_{j \in [k]} P_j \cdot a_j + \sum_{j \in [k]} \sum_{i \in [n]} P_j \cdot M_{j,i} \cdot b_{j,i} \right) \qquad\qquad s.t.$$

$$(a) \; a_j + \sum_{\substack{i \in [n] \\ v_i = 1}} b_{j,i} \geq -w_{\boldsymbol{v},j}^{\boldsymbol{q}} \qquad\qquad \forall \boldsymbol{v} \in \{0,1\}^n, j \in [k]$$

Due to strong-duality, the optimal value of the dual problem is the same of the primal with respect to the fixed assignment $\boldsymbol{w}_q$. By choosing $\boldsymbol{w}_q$ as the minimum over all feasible $\boldsymbol{w}$, we finally obtain the following minimum problem whose optimal value is equal to (20).

$$1 + \min_{\substack{\boldsymbol{a} \in \mathbb{R}^k \\ \boldsymbol{b} \in \mathbb{R}^{k \times n} \\ \boldsymbol{w} \geq 0}} \left( \sum_{j \in [k]} P_j \cdot a_j + \sum_{j \in [k]} \sum_{i \in [n]} P_j \cdot M_{j,i} \cdot b_{j,i} \right) \qquad\qquad s.t. \qquad (21)$$

$$(a) \; a_j + \sum_{\substack{i \in [n] \\ v_i = 1}} b_{j,i} \geq -w_{\boldsymbol{v},j} \qquad\qquad \forall \boldsymbol{v} \in \{0,1\}^n, j \in [k]$$

$$(b) \sum_{j \in [k]} w_{\boldsymbol{v},j} = 1 \qquad\qquad \forall \boldsymbol{v} \in \{0,1\}^n$$

This minimization problem is easily solved as a Linear Programming with $O(k \cdot 2^n)$ variables and constraints. We choose $\boldsymbol{W}^*$ as the optimal solution $\boldsymbol{w}^*$ of the minimum (21).

## C    Approximation of the Lower Bound

In this section of the Appendix, we show a computationally efficient method to compute a lower bound to the value $Q$ in a multiclass classification setting, i.e. $k \geq 2$. We build on the results of Section 4.2, and we will approximate $Q$ by using Theorem 4.2 between properly chosen pairs of the $k$ classes. Consider a weighted, undirected complete graph $G$. Each vertex of the graph represents a class $j \in [k]$, and the edge $\{j, j'\}$ between classes $j, j' \in [k]$, $j \neq j'$, has weight $w_{\{j,j'\}} = \frac{1}{2} \left( 1 - \max_{i \in [n]} |A_{ji} - A_{j'i}| \right)$ computed as in Theorem 4.2. A matching $M$ is a subset of edges such that no two edges of $M$ share an endpoint, i.e. for each $e, e' \in M$, $e \neq e'$, we have that $e \cap e' = \emptyset$. The weight of a matching $M$ is defined as the sum $\sum_{e \in M} w_e$ of the weights of the edges of $M$. The following theorem relates the weight of a matching to the value $Q$.

**Theorem C.1.** *Let $M$ be a matching of $G$, and $Q$ be computed as in (7). Then, $Q \geq \frac{2}{k} \sum_{e \in M} w_e$.*

*Proof.* For each edge $\{j, j'\} = e \in M$, consider the matrix $\boldsymbol{A}^e$ obtained by selecting the two rows of the classes $j$ and $j'$. Let $p^e$ be the PMF over $\{0,1\}^n \times \{j, j'\}$ that achieves the maximum of Theorem 4.2. That is, $p^e$ is the adversarial distribution if we were to only distinguish between the two balanced classes $j$ and $j'$ assuming that we need to also satisfy the constraints imposed by $\boldsymbol{A}^e$.

Let $\boldsymbol{C} = [k] \setminus (\cup_{e \in M} M)$ be the set of classes that do not belong to any edge of the matching $M$. For any $c \in \mathcal{C}$, we let $p^c$ be an arbitrary PMF over $\{0,1\}^n \times \{c\}$ that satisfies the constraints imposed by the $c$ row of the matrix $\boldsymbol{A}$. We give a simple example of such a PMF $p^c$, assuming independence between the attributes. For each $\boldsymbol{v} \in \{0,1\}^n$, we let

$$ p^c(\boldsymbol{v}, c) = \prod_{i \in [n]} A_{c,i}^{v_i} \prod_{i \in [n]} (1 - A_{c,i})^{1 - v_i} \ , $$

and it is easy to verify that this PMF satisfies the constraints imposed by the row $c$ of matrix $\boldsymbol{A}$.

Based on the previous PMFs, we define a PMF $\tilde{p} \in \mathcal{P}(\boldsymbol{A})$ over $\{0,1\}^n \times [k]$. For each $\boldsymbol{v} \in \{0,1\}^n$ and $j \in [k]$, we let

$$ \tilde{p}(\boldsymbol{v}, j) = \begin{cases} \frac{1}{k} p^j(\boldsymbol{v}, j) & \text{if } j \in C \\ \frac{2}{k} p^e(\boldsymbol{v}, j) & \text{for } e \in M : j \in e \end{cases} $$

Observe that this PMF is well defined, as each class is either in $C$ or it belongs to a unique edge in the matching $M$. Moreover, by construction of $\tilde{p}$, the classes are balanced and they satisfy the constraints imposed by matrix $\boldsymbol{A}$.

For each $\{j, j'\} = e \in M$, by construction we have that $1 - \sum_{\boldsymbol{v}} \max(p^e(\boldsymbol{v}, j), p^e(\boldsymbol{v}, j')) = \sum_{\boldsymbol{v}} \min(p^e(\boldsymbol{v}, j), p^e(\boldsymbol{v}, j')) = w_e$ (as $p_e$ achieves the maximum of Theorem 4.2). We have that:

$$ Q \geq \min_{g \in \mathcal{G}} \varepsilon(g, \tilde{p}) = 1 - \sum_{\boldsymbol{v}} \max_{j \in [k]} \tilde{p}(\boldsymbol{v}, j) $$

$$ \geq \sum_{\{j,j'\}=e \in M} \sum_{\boldsymbol{v}} \min \left( \tilde{p}(\boldsymbol{v}, j), \tilde{p}(\boldsymbol{v}, j') \right) $$

$$ = \frac{2}{k} \sum_{\{j,j'\}=e \in M} \sum_{\boldsymbol{v}} \min \left( p^e(\boldsymbol{v}, j), p^e(\boldsymbol{v}, j') \right) $$

$$ = \frac{2}{k} \sum_{e \in M} w_e $$

The first inequality is true because $M$ is a matching, so no two edges of $M$ share an endpoint, and the second equality is due to the definition of $\tilde{p}$. □

In order to maximize the lower bound provided by Theorem C.1, we want to find a matching of $G$ with maximum weight. This optimization problem can be solved in $O(k^3)$ time by using an optimized version of the blossom algorithm (Edmonds, 1965; Lawler, 2001).

# D  Experimental details

In this section, we provide additional details for the experiments in Section 5.

## D.1  Data

We choose the following four datasets with attributes that are widely used benchmarks in ZSL.

**Animals with Attributes 2** (AwA2) consists of 37,322 images of 50 animal classes that are split into 40 seen and 10 unseen classes (Xian et al., 2018a). The dataset contains 85 attributes. We normalize the provided continuous-valued class-attribute matrix, whose entries indicate the strength of the class-attribute association, which we interpret as a probability. We use this matrix as class-attribute matrix. We use the provided binary class-attribute matrix to infer image-level attribute representation for each image to learn the attribute detectors (Appendix D.2).

**aPascal-aYahoo** (aPY) consists of 15,339 images of 32 classes of animals and means of transportation, that are split into 20 seen and 12 unseen classes (Farhadi et al., 2009). Each image is annotated with 64 attributes.

**Caltech-UCSD Birds-200-2011** (CUB) consists of 11,788 images of 200 fine-grained birds classes that are split into 150 seen and 50 unseen classes (Wah et al., 2011). Each image is annotated with 312 attributes.

**SUN attribute database** (SUN) consists of 14,340 images of 717 scenes, e.g., ballroom and auditorium, that are split into 645 seen and 72 unseen classes (Patterson et al., 2014). Each image is annotated with 102 attributes.

For each image, both the SUN and CUB datasets provide multiple crowdsourced attribute annotations. We average such annotations, and we obtain a continuous attribute-representation of each images. For our purposes, i.e., the training of the attribute detectors (Appendix D.2), we round the value of each attribute.

For all these datasets, we use the split between seen and unseen classes suggested by Xian et al. (2018a). Except for AwA2, we obtain the class-attribute matrices by averaging the attribute representation of the images of each class. This is the same strategy used by Romera-Paredes & Torr (2015) in their experiments. For each dataset, we use a pre-trained ResNet-101 (He et al., 2016) as an encoder to extract features from the images. The features are $2048$-dimensional, and they are used as input for the ZSL models.

**Synthetic Data Generation**. In Section 5.2, we generate adversarial synthetic data based on a input class-attribute matrix $\boldsymbol{A} \in [0, 1]^{k \times n}$. To this end, we compute the solution of the Linear Program presented in Section 4.1. The values of the variables $q_{\boldsymbol{v},j}$ that achieve the minimum value of the Linear Program denote an adversarial distribution over the attributes and (balanced) classes that satisfy the constraints imposed by the class-attribute matrix. We remind that $q_{\boldsymbol{v},j}$ denotes the probability that an image has attribute representation $\boldsymbol{v}$ and it belongs to class $j$. We sample classification items from this distribution as follows. We sample a class uniformly at random among the $[k]$ classes, and then we sample a feature vector $\boldsymbol{x}$ with probability $k \cdot q_{\boldsymbol{x},j}$ with $\boldsymbol{x} \in \{0,1\}^n$ (That is, the feature representation is equal to the attribute representation). It is clear that data sampled in this way satisfy the constraints imposed by the class-attribute matrix with attribute functions $\psi_i(\boldsymbol{x}) = x_i$ for $i \in [n]$.

## D.2 Learning Attribute Detectors

For DAP, we need to learn an attribute detector for each attribute. The attribute detectors are classifiers that given an image output either $1$ or $0$, if the attribute appears in the image or not, respectively. We learn the attribute detectors in a supervised fashion on the seen classes, by using the attribute annotations of the images. In AwA2, the attributes are not explicitly annotated for each image, and we use the discrete attribute description of the image's class.

## D.3 ZSL models and training details

In our experiments, we compare the lower bound on the error with multiple ZSL methods. Here, we provide details about the methods how we train them.

**DAP** (Lampert et al., 2014). This method is the first attribute-based method to solve the ZSL problem in the visual domain. It uses attribute detectors trained on the seen classes, and then uses the class-attribute matrix to infer the a posteriori most-probable unseen class. DAP unrealistically assumes attribute independence. We train attribute detectors as explained in the previous section. As suggested by Lampert et al. (2014), we use a uniform prior on the unseen classes. The implementation of DAP is based on the code released by Lampert et al. (2014)[2] under the MIT License[3].

ESZSL (Romera-Paredes & Torr, 2015), SAE (Kodirov et al., 2017), ALE (Akata et al., 2016), and SJE (Akata et al., 2015) learn bilinear maps from image features to the the rows of the class-attribute matrix. At training, they use the class-attribute matrix of the seen classes, while for predictions they use the one of the unseen classes. These methods differ in the definition of the learning objective and the optimization method. In particular, ESZSL and SAE have closed form solutions.

---

[2] https://github.com/zhanxyz/Animals_with_Attributes
[3] https://opensource.org/licenses/MIT

| Method | aPY | AwA2 | CUB | SUN |
|---|---|---|---|---|
| DAP (Lampert et al., 2014) | 30.32 | 40.44 | 27.99 | 19.65 |
| ESZSL (Romera-Paredes & Torr, 2015) | 38.56 | 54.82 | 53.95 | 55.69 |
| SAE (Kodirov et al., 2017) | 16.49 | 58.89 | 46.71 | 59.86 |
| ALE (Akata et al., 2016) | $33.52 \pm 0.35$ | $52.78 \pm 2.78$ | $51.38 \pm 0.77$ | $61.69 \pm 0.40$ |
| SJE (Akata et al., 2015) | $31.93 \pm 0.41$ | $69.17 \pm 1.89$ | $52.23 \pm 0.19$ | $52.94 \pm 0.70$ |
| DAZLE (Huynh & Elhamifar, 2020) | $31.46 \pm 1.52$ | $67.57 \pm 1.33$ | $57.23 \pm 0.70$ | $56.15 \pm 0.57$ |

Table 1: We report the Top-1 balanced average unseen class-accuracy, and standard errors over 5 seeds, for popular attribute-based ZSL. All the metrics are obtained using the splits proposed in Xian et al. (2018a). ESZSL, SAE and DAP do not have intervals because have a closed form solution. For SAE we report results from the semantic to the feature space (Kodirov et al. (2017), Section 4.1).

**ESZSL.** The hyperparameters of the model are $\alpha$ and $\gamma$, which are the regularizer parameter for feature space and the regularizer parameter for the attributes space, respectively. The parameters $\alpha$ and $\gamma$ for each dataset are set as follows: aPY, $\alpha = 3$ and $\gamma = -1$; AwA2, $\alpha = 3$ and $\gamma = 0$; CUB, $\alpha = 3$ and $\gamma = -1$; SUN, $\alpha = 3$ and $\gamma = 2$.

**SAE.** The hyperparameter of the model is $\lambda$ which is a coefficient that controls the trade-off between the decoder and encoder losses. The values of $\lambda$ are set as follows for each dataset: aPY, $\lambda = 4$; AwA2, $\lambda = 0.2$; CUB, $\lambda = 0.2$; SUN, $\lambda = 0.16$.

**ALE.** The hyperparameters of the models are the normalization strategy applied to the class-attribute matrix, and the SGD learning rate $\gamma$. For each dataset, the normalization strategy and the learning rates are: aPY, $\ell_2$ and 0.04; AwA2, $\ell_2$ and $\gamma = 0.01$; CUB: $\ell_2$ and $\gamma = 0.3$; SUN, $\ell_2$ and $\gamma = 0.1$.

**SJE.** The hyperparameters of the model are the normalization strategy applied to the class-attribute matrix, the SGD learning rate $\gamma$, and the margin $m$ for the optimization of the objective. For each dataset we report these parameter, in order: aPY, no normalization, $\gamma = 0.01$, and $m = 1.5$; AwA2, $\ell_2$, $\gamma = 1$, and $m = 2.5$; CUB, mean-centering, $\gamma = 0.1$, and $m = 4$; SUN, mean-centering, $\gamma = 1$, and $m = 2$.

The chosen hyperparameters maximize the balanced accuracy on the validation classes, and lead to test errors on the unseen classes comparable with the benchmarks by Xian et al. (2018a). The implementations of ESZSL, SAE, ALE, SJE are based on a public code repository[4] released under the MIT License. In Table 1, we report the balanced accuracy of each method when trained on the whole set of attributes and seen classes. Interestingly, we note that the accuracy of the methods on aPY is comparable to the accuracy that the methods achieve by only using 15 attributes (Figure 1).

The last ZSL method we consider is DAZLE (Huynh & Elhamifar, 2020) which (1) uses dense attribute-based attention to find local discriminative regions, and (2) embeds each attribute-based feature with the attribute semantic description. The implementation of DAZLE is based on the code released by Huynh & Elhamifar (2020)[5] under the MIT License.

**DAZLE.** The hyperparameters of the model the weight of the the self-calibration loss $\lambda$, the learning rate $\gamma$, the weight decay $w$, and momentum $m$. For each dataset we report these parameter, in order: aPY, $\lambda = 0.1$, $\gamma = 0.0001$, $w = 0.0001$, and $m = 0$; AwA2, $\lambda = 0.1$, $\gamma = 0.0001$, $w = 0.0001$, and $m = 0$; CUB, $\lambda = 0.1$, $\gamma = 0.0001$, $w = 0.0001$, and $m = 0.9$; SUN, $\lambda = 0.1$, $\gamma = 0.0001$, $w = 0.0001$, and $m = 0.9$. We used the same setting as in the released implementation of the model.

**Resources.** We run the experiments on an internal cluster. Most of the methods are executed on CPUs, while for DAZLE we used a GPU NVIDIA GeForce RTX 3090.

---

[4]`https://github.com/mvp18/Popular-ZSL-Algorithms`
[5]`https://github.com/hbdat/cvpr20_DAZLE`

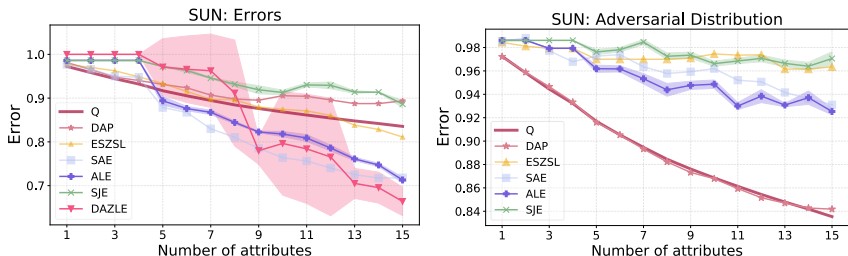

Figure 3: **Comparison of the lower bound with the empirical error.** We plot the lower bound on the error (**Q**), and the error of ZSL methods with attributes (**DAP, ESZSL, SAE, ALE**, and **DAZLE**). The first column reports these values computed on the unseen classes of SUN dataset, varying the number of available attributes. The second column reports the values for the adversarially generated synthetic data. The bands indicate the standard errors on five runs with different seeds for randomized methods.

# E  Additional Experimental Results

## E.1  Additional Results for Section 5.2

In Figure 3, we report the results for the experiments of Section 5.2 for the SUN dataset. The experiments are consistent with our findings. We observe that for the experiments on the SUN data, we still observe that the empirical error of ZSL models roughly follows the trend of the lower bound. This suggests that the lower bound is able to capture how the additional information provided by an attribute leads to improvements of the ZSL models. Moreover, for the adversarially generated synthetic data, we observe that no method is able to achieve errors lower than the lower bound, consistently with our theory.

In this subsection, we also report the empirical results for a method that we call **APA** (Adversarial Predicted Attributes) across all four datasets. APA is an adversarial algorithm that uses a map from attributes to classes that satisfies Theorem 4.3, computed as in Appendix B. The method uses attribute detectors trained on the seen classes (Appendix D.2), and predicts the target classes according to the output of those detectors, as specified in Section 4.3. APA is similar to DAP, except in how the attribute detectors are used to predict the target classes. In Figure 4, we report the result of the experiments for the method APA. We point out that the performance of APA is competitive to the other ZSL approaches, at least using a reduced number of attributes. We remark that this comparison is out of the scope of this paper, but we believe these results open promising direction for further development of adversarial attribute-based zero-shot learning models.

## E.2  Additional Results for Section 5.3

In this subsection, we extend the experiments of Section 5.3 and we propose a way to analytically quantify the similarity between the pairwise lower bound error matrix $L$ and the misclassification matrices. Suppose that a model makes $m$ errors $E = \{(j_1, j_1'), \ldots, (j_m, j_m')\}$ on the data, where for each $i \in [m]$, the pair $(j_i, j_i')$ represents an instance where the ZSL model outputs $j_i$ but the true class is $j_i' \neq j_i$. We compute the ratio between the empirical expected weight of the errors $E$ according to the graph $G$ and the expected weight of errors made uniformly at random between the classes, i.e. $\left(\frac{1}{m} \sum_{(j,j') \in E} w_{j,j'}\right) / \left(\frac{1}{k(k-1)} \sum_{j \neq j'} w_{j,j'}\right)$. We name this quantity *skeweness* (Sk), and we observe that if the ratio is greater than 1, then the misclassification errors $E$ of the ZSL models are skewed towards pair of classes that have larger values in $L$, i.e., they are hard to distinguish. In Table 2, we report the skewness scores computed for all the combination of ZSL models and datasets. We observe that all these quantities are greater than 1. As noted before, this shows that the errors are skewed towards those indicated by our theoretical analysis. We can observe that the skewness is approximately 1 for SAE on the aPY dataset. This is not surprising, as the model has very low performance (16.49% accuracy, see Table 1 in Appendix D) on this ZSL task. We point out that it is very challenging to define a pairwise metric between the entries of $L$ and the misclassification matrix $M$ to describe their similarity. A pairwise metric would fail to capture more complex relations between classes. For instance, consider the scenario where three classes are very

| Method | aPY | AwA2 | CUB | SUN |
|--------|-----|------|-----|-----|
| DAP | $3.65 \pm 0.04$ | $3.44 \pm 0.04$ | $2.96 \pm 0.01$ | $3.09 \pm 0.00$ |
| ESZSL | $3.94 \pm 0.07$ | $3.11 \pm 0.03$ | $3.38 \pm 0.02$ | $3.45 \pm 0.00$ |
| SAE | $1.20 \pm 0.01$ | $3.33 \pm 0.02$ | $3.65 \pm 0.02$ | $3.55 \pm 0.00$ |
| SJE | $5.09 \pm 0.04$ | $3.37 \pm 0.03$ | $3.32 \pm 0.02$ | $3.44 \pm 0.00$ |
| ALE | $4.89 \pm 0.04$ | $3.56 \pm 0.02$ | $3.68 \pm 0.01$ | $3.59 \pm 0.00$ |
| DAZLE | $3.93 \pm 0.05$ | $3.07 \pm 0.05$ | $3.87 \pm 0.01$ | $3.51 \pm 0.00$ |

Table 2: We report for each model the average *skewness* and standard deviation over class-balanced test sets. A value greater than 1 indicates that the model's misclassification error is skewed toward pairs of classes with large values in $\boldsymbol{L}$.

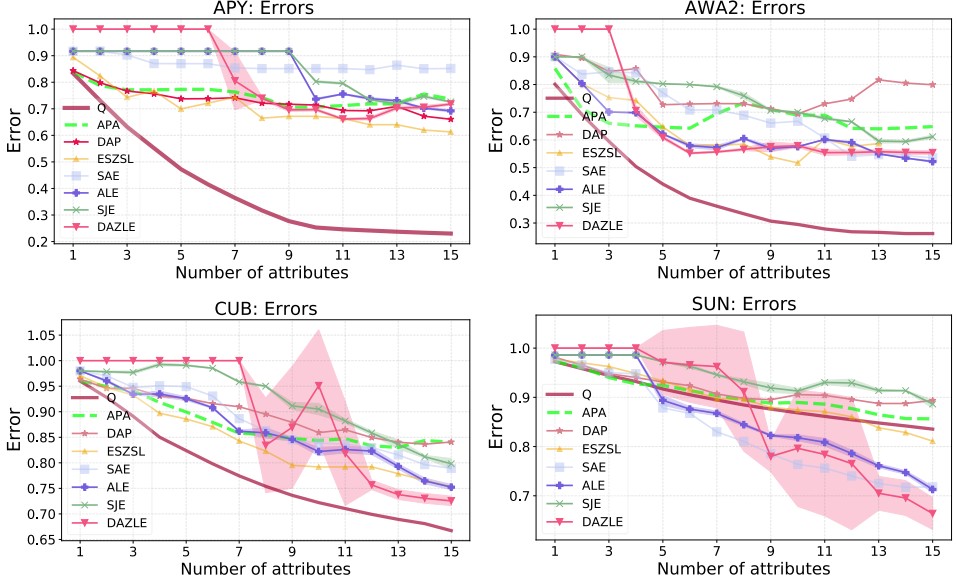

Figure 4: **Comparison of the lower bound on the error with the ZSL models error on the validation classes.** We plot the lower bound on the error ($\mathbf{Q}$) and compare it to the error rates of the ZSL adversarial algorithm (**APA**) and the other ZSL models with attribute (**DAP, ESZSL, SAE, ALE**, and **DAZLE**). The bands indicate the standard error on five runs with different seeds.

hard to distinguish according to the values of their lower bound in $\boldsymbol{L}$. A ZSL model could fail to distinguish between them, and it could always output the same class given an image of any of those three classes. This would imply that we will observe zero misclassifications between a pair of these two classes in the matrix $\boldsymbol{M}$, which is different from the same entry in $\boldsymbol{L}$. Contrarily, the skewness metric is not affected by this problem.

**Additonal Details.** As our lower bound is computed assuming balanced classes, we ensure this assumption holds by sampling the test data uniformly among the unseen classes. In Table 2, we report the skewness averaged on 10 different randomly selected subsets of test data, and the respective standard deviation. Specifically, for each class we sample a number of images equal to the minimum class-size among the unseen classes.

# F   Extension to Incomplete Class-Attribute Information

In this paper, we assume that we are provided a class-attribute matrix $A \in [0,1]^{k \times n}$ such that for each $i \in [n]$ and $j \in [k]$, the entry $A_{j,i}$ provides the probability that we observe attribute $i$ (i.e., $\psi_i(x) = 1$) given that we sample an element of class $j$ (i.e., $y(x) = j$). Formally, the entries of the class-attribute matrix follow equation (1)

$$A_{j,i} = \mathop{\mathbb{P}}_{x \sim \mathcal{D}}[\psi_i(x) = 1 | y(x) = j] \ .$$

In some Zero-Shot Learning problems (Jayaraman & Grauman, 2014; Wang et al., 2017), we are provided a *incomplete* class-attribute matrix. That is, for each class, we are provided reliable information only for a subset of the $n$ attributes. Formally, we are given a matrix $A \in ([0,1] \cup \{*\})^{k \times n}$, where the symbol '$*$' denotes a lack of information. That is, if $A_{j,i} = *$, then the probability of observing attribute $i$ given a sample of an element of class $j$ is arbitrary. Conversely, if $A_{j,i} \in [0,1]$, then we are provided the same information considered in the original setting, and we know that the relation between attribute $i$ and class $j$ follows equation (1).

We can easily extend the lower bound developed in Section 4 for incomplete class-attribute matrices. In fact, in the original formulation in the paper, each entry of the class-attribute matrix defines a constraint on the joint distribution $p$ over classes and attributes. If we are not given a relation between class $j$ and attribute $i$, i.e. $A_{j,i} = *$, then we simply do not specify that constraint. The lower bound can still be computed by using a Linear Program as in Section 4.1. In the case of incomplete class-attribute matrix, we specify the constraints $(a)$ of the Linear Program only for pairs of attribute $i$ and class $j$ such that $A_{j,i} \in [0,1]$.