# OpenReview forum: "Tight Lower Bounds on Worst-Case Guarantees for Zero-Shot Learning with Attributes"
_NeurIPS.cc/2022/Conference — NeurIPS 2022 Accept_

### Official Review · Reviewer_oqVB · 2022-07-10

**Rating:** 6
**Confidence:** 3
**Soundness:** 3 good
**Presentation:** 3 good
**Contribution:** 3 good

**Summary:**

Authors derive a new bound for zero-shot learning (ZSL) corresponding to the worst-case Bayes Error, taken across all distributions that satisfy some given attribute-class matrix. The bound assumes access to a perfect item-to-attribute mapping. Authors show that computing the derived bound amounts to solving an LP problem. Authors show that this problem admits a closed-form solution in the binary case (given along with the expression of the actual classifier that achieves that bound), and can be approximated in the multi-class setting. They also show that the bound is tight in general.

**Questions:**

### Interpreting experimental results

The first row of Fig.1 is quite hard to interpret, because a large part of the gap between the lower bound and methods seems to be due to the poor item-to-attribute mapping generalization. I find the set of adversarial experiments more interesting. However, some methods do go below the lower bound, which lets me think that the actual "best" classifier could further 'break' the theoretical lower bound. Is this due to sampling / approximate computation of the bound in the multi-class setting ? Could authors clarify ?

### Motivating the use of worst-case bounds

I believe I'm not super clear with the motivation of using worst-case bounds for ZSL. The bound considers the performance of the best possible classifier in the worst-case distribution. However in practice, how likely is this to happen (such that the bound actually applies) ? The results on SUN dataset for instance, clearly show that the lower bound is way above most methods (even with the presence of training/testing shift), and the gap keeps increasing as the number of attributes increases. Would best-case lower bounds be trivial here? i.e. given some $\textbf{A}$, how well could any method achieve, assuming the 'easiest' distribution compatible with $\textbf{A}$ . I'd be curious to hear authors' thoughts.

### Sanity checking

Binary results would have been easier to sanity check / interpret, because less confounding factors. In particular, given that the found value Q neither depends on a particular distribution or on the number of attributes, there would be no need to adversarially choose the distribution or to worry about approximating Q, and a direct apple-to-apple comparison would have been possible. Is there any particular reason for not experimenting in a binary situation ?





**Ethics Review Area:**

["I don’t know"]

**Limitations:**

Authors do discuss that the bound assumes perfect item-to-item mapping, which is far from true in real life. Authors also discuss the difficulty associated with computing the bound (scaling to multi-class and to multiple attributes).

**Strengths And Weaknesses:**

## Strength

- The paper is well written. As a non-expert for ZSL, I can say the notations and ideas were well introduced, and easy to follow. The proper amount of background on the problem was introduced.

- The general problem of formally characterizing the difficulty of a zero-shot problem is an interesting one.

- The work appears substantial to me. The formalization of the problem, the bound/its computation and the tightness results seem non-trivial and novel.

- The result in the binary case appears non-trivial and could already be of practical relevance. If I understand correctly, authors provide a closed-form expression of a classifier $g_a(\textbf{v})$ that is essentially optimal, independently of the distribution, i.e optimal $\forall ~ p, ~ p \in \mathcal{P}(\textbf{A})$. The classifier has a simple expresison that can be readily computed from the class-attribute matrix $\textbf{A}$.

## Weaknesses (c.f Questions below)

C.f questions below

---

> ### Author Response · Authors · 2022-08-02
> **Rebuttal**
>
> Thank you for taking the time necessary to review our paper, and for all the helpful comments and feedback.
>
> ---
>
> **Question**: The first row of Fig.1 is quite hard to interpret, because a large part of the gap between the lower bound and methods seems to be due to the poor item-to-attribute mapping generalization. I find the set of adversarial experiments more interesting. However, some methods do go below the lower bound, which lets me think that the actual "best" classifier could further 'break' the theoretical lower bound. Is this due to sampling / approximate computation of the bound in the multi-class setting ? Could authors clarify ?
>
> **Response**: This is indeed due to the approximation of evaluating the error of the methods using a finite sample.
>
> ---
>
> **Question**: Motivating the use of worst-case bounds. I believe I'm not super clear with the motivation of using worst-case bounds for ZSL. The bound considers the performance of the best possible classifier in the worst-case distribution. However in practice, how likely is this to happen (such that the bound actually applies) ? The results on SUN dataset for instance, clearly show that the lower bound is way above most methods (even with the presence of training/testing shift), and the gap keeps increasing as the number of attributes increases. Would best-case lower bounds be trivial here? i.e. given some , how well could any method achieve, assuming the 'easiest' distribution compatible with  . I'd be curious to hear authors' thoughts.
>
> **Response**: We are using a worst-case analysis as we would like to quantify the quality of the information that we are given. As the reviewer observes, the error is a function of the joint distribution of attributes and classes. However, we cannot estimate properties of that distribution as we do not have access to any training points. The only information we are given is the class-attribute matrix, and there are many distributions that conforms with the constraints imposed by this matrix. Therefore, our lower bound is with respect to the worst-case distribution: without additional assumptions, no algorithm can guarantee to do better.
>
> The error of the best map from attributes to classes in a best-case distribution could be significantly lower. This is also described in the Example starting at line 209. In this case, even with two classes having the same identical attribute description, the error of the best-case distribution is 0.
>
> ---
>
> **Question**: Binary results would have been easier to sanity check / interpret, because less confounding factors. In particular, given that the found value Q neither depends on a particular distribution or on the number of attributes, there would be no need to adversarially choose the distribution or to worry about approximating Q, and a direct apple-to-apple comparison would have been possible. Is there any particular reason for not experimenting in a binary situation ?
>
> **Response**: Thanks, we focused on multi-class just because popular ZSL datasets are multi-class. Please also note that the closed formula for the binary setting is hiding the fact that the value Q is still computed by choosing the worst-case distribution (the Theorem 4.2 shows a closed formula for the solution of the LP in a binary setting).
>
> ---
>
> **Question**: Authors do discuss that the bound assumes perfect item-to-item mapping, which is far from true in real life.
>
> **Response**: That's right, but in addition to the discussion in the paper of this limitation, we'd like to reiterate that the perfect item-to-attribute mapping assumption is used to focus on the error due to the map from attribute to classes. In our work we want to quantify the error due to the quality of the information provided for the unseen classes - the class attribute matrix. There are, of course, other  sources of errors, such as domain shift, that are not unique to ZSL. These other errors are additional to the error computed here, and of course the lower bound still holds even if we consider other sources of error.

---

### Official Review · Reviewer_9VLD · 2022-07-11

**Rating:** 7
**Confidence:** 3
**Soundness:** 4 excellent
**Presentation:** 3 good
**Contribution:** 3 good

**Summary:**

This paper studies the generalization of zero-shot learners with access to attributes. In this work, zero-shot learning is decomposed into two stages as is typically done in practice. The first stage consists of learning to extract attributes from input data. In the second stage, new classes are presented and must be classified using the attribute information that was extracted during the first stage. The results presented in this paper are applicable to the second stage under the assumption that attribute extraction can be learned perfectly.

The authors derive the Bayes optimal classifier for this second stage in terms of an unknown joint distribution over attributes and labels. By considering constraints on this joint distribution in terms of the class-attribute matrix, they are able to produce a lower-bound, $Q$, on the error of a Bayes optimal classifier. A linear program is then designed whose optimal solution corresponds to this $1-Q$. They solve this LP for balanced binary classification and show that the lower-bound is tight in this case.

The authors then show that this lower bound is tight in general by showing that a randomized attribute-class classifier can be designed whose error is upper-bounded by the lower bound. Moreover, this classifier can be computed as the solution of a linear program.

The empirical analysis confirms the theoretical findings on a number of standard ZSL classification problems. Two sets of experiments are conducted: first using the standard ZSL setup, and second using synthetic attribute labels that are generated to match the class-attribute matrix for each task. The lower bound is respected in both cases.

**Post-rebuttal:** Following the author's response, I have increased my score from Weak Accept to Accept. After reviewing the literature referenced by the authors, I am in agreement that the results are a great contribution than I thought them to be initially.

**Questions:**

My main concern with this work is the extent to which the results generalize to the true ZSL setting where there is a distribution shift and the attribute functions must be learned. I am also concerned that the results presented may already be known within the boolean function learning literature --- I hope that another reviewer has greater expertise in this area or that the authors themselves can comment on this comparison.

The authors write that "this result also formally proves that for binary classification, the worst-case is determined by a single attribute, and there is no compounded benefit in having multiple attributes". This statement is entirely dependent on the fact that all attribute functions are assumed to be given to the classifier. In a true ZSL setting, where these attribute functions must be learned, some attributes may be easier to recover than others and so we would certainly expect a compound effect from having access to multiple attributes.

### Minor comments

Minor comments:

I personally find the concatenation notation to be confusing as it could be mistaken for a product. Perhaps consider adopting `[v, v']`, `cat(v, v')` or similar.

Some extra structure (e.g. paragraph headings) in the preliminary chapter could make things easier to parse.

Before Equation 3, I would have found it useful to include a brief explanation in words of the optimal binary classifier. Something like: "In words, the optimal classifier chooses the class which is most likely under the given attribute configuration."

- L255: "A[n] attribute-class classifier"

**Limitations:**

The authors provide a thorough and balanced discussion of the limitations of their work. Here I include some additional minor suggestions.

The bounds presented are of the maximum of the minimum and not the more common minimax risk. The latter is more common in learning theory, as we want to know the performance of the best *learning* algorithm on the hardest problem we consider (thus the learner is problem-agnostic). The bound presented ignores learning and lower-bounds the best performance of a perfect classifier with all information of the task. In this case, by von Neumann's minimax theorem the two are equivalent (as is used in the proofs in the appendix). Perhaps this could be further discussed.

Assuming access to the attribute functions is a strong assumption that moves away from the core of ZSL. The first stage of ZSL is typically to recover this attribute mapping. This two-stage decomposition is represented accurately in the fourth paragraph of the introduction, in the preliminaries. This limitation of the work is discussed in Section 6. It would be valuable to discuss this limitation in terms of the theoretical results that are presented too. What needs to be done to address both stages of learning? To what extent can errors in the attribute functions be incorporated into the proofs? Where does this fail?

**Strengths And Weaknesses:**

## Strengths

ZSL is a particularly difficult area to establish theoretical results in. To my knowledge, the results presented in this work are novel and make progress towards understanding theoretical limitations of ZSL.

The empirical analysis is practically relevant and thorough. The authors evaluate their theoretical results on a good range of ZSL tasks. Further, they actively work towards matching the settings with their theoretical assumptions in their second set of experiments.

## Weaknesses

Results are interesting albeit simple in nature. The authors assume that the first stage of learning has been absolutely successful and the attribute functions have been recovered. They then analyze the worst-case error for the second phase where a classifier is learned from the attributes to classes. This is interesting and valuable but it excludes one of the key tenets of zero-shot learning: that some classes have no instances available at training time. This is addressed in the first phase of learning and is, arguably, the more interesting part of the ZSL paradigm. It is possible that the authors analysis can play a role in understanding the full ZSL pipeline, but no effort is made to understand this and I expect that the linear programs are too rigid to account for the necessary changes.

The authors are essentially studying classification with boolean functions with an additional distributional assumption. This is not an area of the literature that I know exceedingly well, but I would expect a discussion from the authors within their paper. Even initial work in this setting [1] capture the binary version of the classification problem that the authors consider.


[1] A theory of the learnable. L. Valiant. Artificial Intelligence and Language Processing, 1984.

---

> ### Author Response · Authors · 2022-08-02
> **Rebuttal (1/2)**
>
> We would like to thank you for the careful review of our paper. We appreciate the insightful feedback and suggestions.
>
> ---
>
> **Question**: Results are interesting albeit simple in nature. The authors assume that the first stage of learning has been absolutely successful and the attribute functions have been recovered. They then analyze the worst-case error for the second phase where a classifier is learned from the attributes to classes. This is interesting and valuable but it excludes one of the key tenets of zero-shot learning: that some classes have no instances available at training time. This is addressed in the first phase of learning and is, arguably, the more interesting part of the ZSL paradigm. It is possible that the author's analysis can play a role in understanding the full ZSL pipeline, but no effort is made to understand this and I expect that the linear programs are too rigid to account for the necessary changes.
>
> **Question**: My main concern with this work is the extent to which the results generalize to the true ZSL setting where there is a distribution shift and the attribute functions must be learned.
>
> **Question**: Assuming access to the attribute functions is a strong assumption that moves away from the core of ZSL. The first stage of ZSL is typically to recover this attribute mapping. This two-stage decomposition is represented accurately in the fourth paragraph of the introduction, in the preliminaries. This limitation of the work is discussed in Section 6. It would be valuable to discuss this limitation in terms of the theoretical results that are presented too. What needs to be done to address both stages of learning? To what extent can errors in the attribute functions be incorporated into the proofs? Where does this fail?
>
> **Response (to the above three related questions)**:
> Thanks for providing this detailed feedback. We would like to remark that we are addressing an open-question first posed in a seminal work of ZSL (Romera-Paredes & Torr, 2015), and we are the first work to provide a non-trivial lower-bound in a ZSL setting. Moreover, our work is able to rigorously and mathematically quantify the quality of the information provided for the unseen classes (the class-attribute matrix). This is noticeable, as this is the only information that we have available in ZSL for the unseen classes.
>
> As we address in more detail in our General Comment, it is very challenging to provide a full mathematical analysis for ZSL. In fact, the attribute functions are learned with respect to the distribution of the data of the seen classes, and then used on the unseen classes. The error due to this distribution shift can be theoretically quantified according to the results of Transfer Learning of Ben-David et al, 2010. However, this error is not computable in a ZSL setting, as no labeled data is available for the unseen classes: in theory, the error due to the distribution shift can be arbitrarily large.
>
> Our lower bound still applies in the true ZSL setting. Indeed, the distribution shift can be a source of additional errors, as the attribute functions are learned with respect to a different distribution. However, this is an additional error that causes noise in retrieving the correct attribute representation. Even if we were able to learn the true attribute functions, no algorithm can guarantee error better than our lower bound (if the only information provided for the unseen classes is the one of the class-attribute matrix).
>
> In order to incorporate the error of the attribute functions, we would need to quantify the error due to the distribution shift. Since no labeled data is available, this error can be arbitrarily large. That is, in theory, we do not know the error rates of the learned attribute functions for the unseen classes. This discussion motivates that is very challenging to characterize the error due to the first stage of learning in ZSL without introducing other assumptions on the distribution of the unseen classes (that we cannot verify, as no labeled data is available for the unseen classes.).
>
> In our work, we provide a lower bound without introducing any further assumption on the distribution of the unseen classes, that is based on the only information available - the class attribute matrix.
>
> ---

---

> > ### Author Response · Authors · 2022-08-02
> > **Rebuttal (2/2)**
> >
> > **Question**: The authors are essentially studying classification with boolean functions with an additional distributional assumption. This is not an area of the literature that I know exceedingly well, but I would expect a discussion from the authors within their paper. Even initial work in this setting [1] capture the binary version of the classification problem that the authors consider.
> > [1] A theory of the learnable. L. Valiant. Artificial Intelligence and Language Processing, 1984.
> >
> > **Question**: I am also concerned that the results presented may already be known within the boolean function learning literature --- I hope that another reviewer has greater expertise in this area or that the authors themselves can comment on this comparison.
> >
> > **Response (to the above two related questions)**:
> > Thanks for raising this point. The Boolean classification learning problem is different from the attribute-based ZSL problem studied in this paper. In the Boolean classification learning problem the goal is to design a boolean function that best conforms to the training set. The error is evaluated with respect to the distribution whose training points are sampled from, and we are interested in the number of samples required (sample complexity) to approximately identify the best boolean function, regardless of what the error of the best Boolean function is.
> >
> > In our problem, we do not have access to samples from our target distribution. Instead, we are only given a set of statistical Boolean relations (the class-attribute matrix), and the question is how good is the description of the classes in terms of these Boolean relations. This is evaluated as the error of the best Boolean function with respect to a worst-case (adversarial) distribution. This question is unique to the ZSL setting.
> >
> > ---
> >
> > **Question**: The authors write that "this result also formally proves that for binary classification, the worst-case is determined by a single attribute, and there is no compounded benefit in having multiple attributes". This statement is entirely dependent on the fact that all attribute functions are assumed to be given to the classifier. In a true ZSL setting, where these attribute functions must be learned, some attributes may be easier to recover than others and so we would certainly expect a compound effect from having access to multiple attributes.
> >
> > **Response**: Thanks for the feedback. This was meant as a simple remark to connect this result to similar work in weak supervision. In the multi-class setting, that is the main focus of our work, this observation is not relevant and there is indeed a benefit in having multiple attributes.
> >
> > To further clarify, the lower bound in the binary case is determined by a single attribute. In the case of additional noise on the attribute detectors, we agree that we could benefit from having access to multiple attributes: however our lower bound still applies, and the noise of the attribute detectors would cause additional error.
> >
> > ---
> >
> > **Question**: The bounds presented are of the maximum of the minimum and not the more common minimax risk. The latter is more common in learning theory, as we want to know the performance of the best learning algorithm on the hardest problem we consider (thus the learner is problem-agnostic). The bound presented ignores learning and lower-bounds the best performance of a perfect classifier with all information of the task. In this case, by von Neumann's minimax theorem the two are equivalent (as is used in the proofs in the appendix). Perhaps this could be further discussed.
> >
> > **Response**: Thanks for the suggestion. As noted by the Reviewer, in our case the values of the mini-max and the maxi-min are equivalent. This fact is used to show that the lower bound is tight.
> >
> >
> > For the presentation of our lower bound, we are interested in the maxi-min. In fact, for each possible distribution (according to the class-attribute matrix), we consider the error of the best classifier with respect to that distribution, and we take the maximum among all these errors as the lower bound. This guarantees that any algorithm cannot guarantee an error lower than this lower bound.
> >
> > The minimax classifier is used to prove that the lower bound is tight (Theorem 4.3 and Appendix B). In fact, in learning theory, the minimax classifier is naturally related to the concept of upper bound as the error of each classifier is computed with respect to the worst-case distribution. By using the fact that min-max = max-min, we show that our lower bound Q (from max-min) is tight, as the minimax classifier (Theorem 4.3) has error at the most Q.

---

> > > ### Comment · Reviewer_9VLD · 2022-08-09
> > > **Response to rebuttal**
> > >
> > > Thank you for your detailed response.
> > >
> > > I feel that the majority of my concerns have been addressed. I apologize for my late response; I needed some time to read over the provided reference [Romera-Paredes & Torr, 2015]. Several reviewers, including myself, were concerned by the relevance of the result to the  "true ZSL" setting. The response has made a good argument that even though the current results are limited, they are 1) still a valuable contribution in addressing an open problem of ZSL and 2) still applicable to the true ZSL setting, even if potentially loose. I am still not entirely convinced by the value of providing an answer to this open question without tackling more components of ZSL. But I am now more in favour of the author's argument.
> > >
> > > Following the rebuttal, I have increased my score to accept. I agree that the contributions are more significant than I initially understood.

---

### Official Review · Reviewer_Rmpv · 2022-07-11

**Rating:** 6
**Confidence:** 4
**Soundness:** 3 good
**Presentation:** 3 good
**Contribution:** 3 good

**Summary:**

This paper presents a tight lower bound for attribute-based zero-shot learning based on the class-attribute matrix. When compared with previous bound from limit amount of work studying this, the proposed bound is the first non-trivial tight lower bound.  The proposed bound is empirically validated with several methods on popularly used benchmark datasets, and can be also used for pairwise misclassification prediction.

**Questions:**

- Application of the bound, can it be used to improve the zero-shot learning performance? Or come up with new methods?

**Limitations:**

- This bound reveals the importance of class-attribute matrix, which itself is very hard to obtain in real-world application

**Strengths And Weaknesses:**

Strengths
- First right non-trivial bound for attribute-based zero shot learning
- Good empirical study to demonstrate the bound and its application
- Source code provided

Weakness
- How can this bound used for improve zero-shot learning in addition to the pairwise misclassification prediction?
- Any relationship between pairwise misclassification from this method and from confusion matrix? (PS: I know there is no samples for unseen class)

---

> ### Author Response · Authors · 2022-08-02
> **Rebuttal**
>
> Thank you for taking the time to review our paper, and for your insightful comments and questions.
>
> ---
>
> **Question**: How can this bound used for improve zero-shot learning in addition to the pairwise misclassification prediction?
>
> **Question**: Application of the bound, can it be used to improve the zero-shot learning performance? Or come up with new methods?
>
> **Response (to the above two related questions)**: Before improving, it is important to theoretically show the limitations of ZSL, and this is the main scope of our paper. In particular, we show that if the only information provided is the one of the class-attribute matrix, no attribute-based ZSL method can theoretically guarantee an error better than our lower bound. Please also see additional discussion in the general comment.
>
> This framework can motivate several possible extensions:
> 1. If we are given a class-attribute matrix whose value of the lower bound is very large, it could be beneficial to collect additional attributes.
> 2. If we are allowed to modify the task, we could use the information provided to group together classes that are very hard to distinguish between one another. Our method provides a rigorous way to quantify how hard two classes are to distinguish. As an example, if we know it is hard to distinguish between "blue whale" and "white whale", it would be beneficial to group these classes together.
> 3. We could investigate ZSL methods that combine the attributes adversarially (in a minimax sense). This computation is possible within our framework and we perform a preliminary investigation in Appendix E.1. Adversarial methods have shown good results in other ML fields (as weak supervision, e.g., see Mazzetto et al 2021), and this could be an interesting future direction.
>
> ---
>
> **Question**: Any relationship between pairwise misclassification from this method and from confusion matrix? (PS: I know there is no samples for unseen class)
>
> **Response**: In addition to the results in Section 5.3 (Figure 2) and Appendix E.2, it’s interesting to think about whether the relationship can be formalized in an analytical way. It is challenging to define a pairwise relationship between the entries of the pairwise misclassification matrix and the confusion matrix. In fact, the pairwise misclassification matrix provides a pairwise lower bound based on the information available and it only depends on the input class-attribute matrix. Instead, the confusion matrix is different for each possible ZSL model. Following the example in Appendix E.2 (see Line 831, or Line 836 on the revisioned appendix), consider a scenario where three classes are very  hard to distinguish between one another according to the pairwise misclassification matrix. Accordingly, a ZSL model could fail to distinguish between them, and it could always output the same class given an image of any of those three classes. This would imply that we will observe zero misclassifications between a pair of two classes in the confusion matrix.
>
> ---
>
> **Question**: This bound reveals the importance of class-attribute matrix, which itself is very hard to obtain in real-world application.
>
> **Response**: Yes, the class-attribute matrix is central to this well-studied setting, which is why our work to rigorously quantify the quality of the information it contains is significant. As mentioned above, this also points to a potential application of our work: it could guide the work of incrementally creating a class-attribute matrix by quantifying which classes are sufficiently well described and which are not.

---

### Official Review · Reviewer_YtKa · 2022-07-12

**Rating:** 5
**Confidence:** 4
**Soundness:** 3 good
**Presentation:** 3 good
**Contribution:** 2 fair

**Summary:**

This paper studies a problem related to zero-shot learning. It aims to answer the question of what is the worst case performance of a classifier, if only given a binary attribute-class association matrix. The lower bound is expressed as a linear program that computes the maximum correlation between any pair of attributes. The paper presents a tight lower bound that can predict the performance of the best classifier given true attribute information of the inputs. However, there are also concerns that the problem being studied is too simple and constrained.

**Questions:**

- Why would the adversarial distribution generate a better error rate for other methods? Could you give an example of what the adversarial distribution looks like?

- Is it correct to understand that all methods use images as input except the lower bound “Q”?

**Limitations:**

Yes

**Strengths And Weaknesses:**

--------------------------
Strengths

- The paper studies a simplified problem that frequently occurs in zero-shot learning and gives insight on when we expect the model to generalize to unseen classes that are defined by an attribute-class matrix.

- The paper shows that the bound is tight.

- The experiments show that the theory predicts the behavior relatively well, especially on adversarial distributions.

-------------------------
Weaknesses

- The theory in the paper generates an intuitive explanation for binary classification (n=2). However, the linear program in general is hard to give an analytical interpretation.

- Although the paper title mentions ZSL, the paper itself actually does not explicitly deal with ZSL but studies a much simpler form of error bound on using only input dimension statistics (probability of attribute i given class j). There might be a lack of mathematical depth for such a constrained problem.

- The assumption might not be realistic enough. Many problems often give a couple attributes that “define” the class, rather than providing full statistics that span across all attribute dimensions. Many attributes are ones that we “don’t care”, and could be spurious attributes if they are provided to the classifier. For example, most of the zebras are on grass, and they are rarely in water. However, we shouldn’t consider these attributes in the classifier even if p(“grass” | zebra) might be close to 1.0, and p("grass" | other animal) is small. This leads to the spurious feature problem, and the hope is that if we could learn disentangled attributes from the space, then it could be solved by focusing only on causal attributes. Now, with the framework presented here, it would be nice if it could be extended to cases where some entries in A are “don’t care”, which motivates the original formulation of ZSL. This might be a trivial modification to the paper but could make the setup more realistic. In the end, attribute-based ZSL should focus on attributes that define the class instead of attributes that correlate with the class.

---

> ### Author Response · Authors · 2022-08-02
> **Rebuttal (1/2)**
>
> Thank you for your work in reviewing our manuscript, and for providing thorough feedback.
>
> ---
>
> **Question**: “The theory in the paper generates an intuitive explanation for binary classification (n=2). However, the linear program in general is hard to give an analytical interpretation.”
>
> **Response**: As discussed in the General Comment, the main contribution of our work is developing a mathematical method for estimating a lower bound to the error of ZSL  as a function of the information available to the algorithm. As pointed out by the reviewer, this lower bound in a ZSL scheme is not straightforward to estimate and interpret. As in the binary case, the linear program computes the worst-case distribution of the joint distribution of classes and attributes, such that this joint distribution satisfies the information provided by the class-attribute matrix. Intuitively, this worst-case distribution maximizes the overlap of classes that have the same attribute representation (see Line 203). Unfortunately, the setup is too complicated for a simple transparent interpretation.
>
> ---
>
> **Question**: “Although the paper title mentions ZSL, the paper itself actually does not explicitly deal with ZSL but studies a much simpler form of error bound on using only input dimension statistics (probability of attribute i given class j). There might be a lack of mathematical depth for such a constrained problem.”
>
> **Response**: We respectfully disagree with the statement that the work doesn’t deal explicitly with ZSL. A central aspect of ZSL is making predictions given only descriptions of unseen classes. Here, we provide the first rigorous way of quantifying the quality of those descriptions in a classic ZSL setting. There are, of course, other sources of errors, such as domain shift, that are not unique to ZSL. These other errors are additional to the error computed here, and of course the lower bound still holds even if we consider other sources of error.
>
> ---
>
> **Question**: The assumption might not be realistic enough. Many problems often give a couple attributes that “define” the class, rather than providing full statistics that span across all attribute dimensions. Many attributes are ones that we “don’t care”, and could be spurious attributes if they are provided to the classifier [...]. Now, with the framework presented here, it would be nice if it could be extended to cases where some entries in A are “don’t care”, which motivates the original formulation of ZSL. This might be a trivial modification to the paper but could make the setup more realistic. In the end, attribute-based ZSL should focus on attributes that define the class instead of attributes that correlate with the class.
>
> **Response**: We thank you for your suggestion, and we agree with this feedback. This is indeed possible within our proposed framework and it is a trivial extension. In fact, each attribute-class constraint specifies a constraint for the joint distribution of attributes and classes. If a constraint for class j and attribute i is not specified (i.e., the entry i,j of the matrix A has a “don’t care value”), we simply need to remove the constraint with indices i and j from the Linear Program (constraint (a) on page 5).
>
> We also thought about this setting, but we decided to not add it to our final draft because of the relevance with respect to the considered attribute-based methods and datasets. We revisioned the paper and submitted another version with this extension added to the Appendix.
>
> ---
>
> **Question**: Why would the adversarial distribution generate a better error rate for other methods?
>
> **Response**: The synthetic data generated according to the adversarial distribution is different from the data on the real dataset, while they indeed share the same class-attribute matrix, so in general these two error rates are not directly comparable. As explained starting from Line 328, we use the same synthetic data for both training and testing (i.e., the data of seen classes and unseen classes is from the same distribution) in order to minimize the error due to the domain shift, and highlight the error due to the quality of the information provided (class-attribute matrix).
>
> ---

---

> > ### Author Response · Authors · 2022-08-02
> > **Rebuttal (2/2)**
> >
> > **Question**: Could you give an example of what the adversarial distribution looks like?
> >
> > **Response**: The adversarial distribution is generated adversarially based on the class-attribute matrix (see subsection starting on Line 733 of the Appendix for the details - 738 on the revisioned appendix). The synthetic data is generated in such a way that the feature description of each classification item is equal to the attribute description, and it is sampled according to the adversarial joint distribution of classes-attributes (computed from the class-attribute matrix). An example of adversarial joint distribution of class-attribute is given in the Example in Line 209.
> >
> > ---
> >
> > **Question**: Is it correct to understand that all methods use images as input except the lower bound “Q”?
> >
> > **Response**: That’s correct. All methods are run on the image datasets. The lower bound Q is only computed based on the class-attribute matrix. There are additional details on the data used and the training of the models in Appendix D.

---

### Author Response · Authors · 2022-08-02
**General Comment**

We would like to thank the reviewers for their close reading of our paper, and for their insightful comments and feedback.

In addition to individual replies below, we’d like to address a common point of discussion regarding our analysis and why it is a significant step in developing a rigorous understanding of zero-shot learning (ZSL). While ZSL has achieved useful results in practice, there is little theoretical work for ZSL. In this paper, we present the first non-trivial lower bound for ZSL for a seminal category of ZSL methods (attribute-based): our work addresses a 7-years old unresolved open question from the influential work of Romera-Paredes & Torr, 2015. Noticeably, we can rigorously evaluate the quality of the information that is available for the unseen classes.

Our analysis gives a worst-case lower bound for attribute-based to ZSL algorithms, in the sense that no algorithm can guarantee an error better than the lower bound. This is an important tool, as it allows us to quantify the quality of the information provided about unseen classes (the class-attribute matrix) and assess if more information is needed.

A common question regarded the implications of decomposing the sources of errors into those from the quality of the class-attribute matrix and those from domain shift affecting the classification of attributes. In this paper, we provide the first rigorous way of quantifying the error due to the quality of the information provided for the unseen classes - the class-attribute matrix. This is important, as this is the only information that we have available for the unseen classes.
Of course, there is also the error component due to domain shift from the seen classes to the unseen classes. We would like to remark that this error is not unique to ZSL, and it is studied in theoretical work on Transfer Learning (Ben-David et al, 2010). However, it is impossible to estimate this error in a ZSL setting, as no labeled data is available for the unseen classes.
Finally, we would like to remark that while domain shift can indeed be a source of error, this error is additional and our lower bound still holds.

---

### Meta-Review · Area_Chair_rcSt · 2022-08-29

**Recommendation:** Accept
**Confidence:** Certain

**Metareview:**

The paper provides a lower bound on the error attainable by a zero-shot learning method in terms of a linear program involving the class-attribution matrix which is provided as side-information.  All the reviewers agree that the analysis is novel and studies an important problem. The main concerns are that: a. the problem itself is studied in a highly constrained setup, b. hard to understand the exact insights from the result, and how it could be used to further the state of the art.
In particular, it would be important to tone down some of the claims, and clarify to the reader that the claims are in context of the considered problem setup only.

**Award:**

No

---

### Decision · Program_Chairs · 2022-09-14

Accept